# Benzoquinoline Chemical Space: A Helpful Approach in Antibacterial and Anticancer Drug Design

**DOI:** 10.3390/molecules28031069

**Published:** 2023-01-20

**Authors:** Claudiu N. Lungu, Violeta Mangalagiu, Ionel I. Mangalagiu, Mihaela C. Mehedinti

**Affiliations:** 1Department of Surgery, Emergency Country Clinical Hospital, 800010 Galati, Romania; 2Faculty of Chemistry, Alexandru Ioan Cuza University of Iasi, 11 Carol 1st Bvd, 700506 Iasi, Romania; 3Department of Morphological and Functional Science, University of Medicine and Pharmacy, Dunarea de Jos, 800017 Galati, Romania; 4Faculty of Food Engineering, Stefan cel Mare University of Suceava, 13 Universitatii Str., 720229 Suceava, Romania; 5Institute of Interdisciplinary Research-CERNESIM Centre, Alexandru Ioan Cuza University of Iasi, 11 Carol I, 700506 Iasi, Romania

**Keywords:** benzoquinoline, chemical space, pharmacophore, molecular database, antimicrobial, anticancer

## Abstract

Benzoquinolines are used in many drug design projects as starting molecules subject to derivatization. This computational study aims to characterize e benzoquinone drug space to ease future drug design processes based on these molecules. The drug space is composed of all benzoquinones, which are active on topoisomerase II and ATP synthase. Topological, chemical, and bioactivity spaces are explored using computational methodologies based on virtual screening and scaffold hopping and molecular docking, respectively. Topological space is a geometrical space in which the elements composing it can be defined as a set of neighbors (which satisfy a particular axiom). In such space, a chemical space can be defined as the property space spanned by all possible molecules and chemical compounds adhering to a given set of construction principles and boundary conditions. In this chemical space, the potentially pharmacologically active molecules form the bioactivity space. Results show a poly-morphological chemical space that suggests distinct characteristics. The chemical space is correlated with properties such as steric energy, the number of hydrogen bonds, the presence of halogen atoms, and membrane permeability-related properties. Lastly, novel chemical compounds (such as oxadiazole methybenzamide and floro methylcyclohexane diene) with drug-like potential, active on TOPO II and ATP synthase have been identified.

## 1. Introduction

Bioactivity, or biological activity, outlines the property of a stimulus (radiation, molecule, biomolecule, drug) to evoke a biological response from living matter [1,2]. In pharmacology, this activity is determined by ligand molecule pharmacophores, constituting and shaping the interactions between the ligand molecule and its target (i.e., receptor) [3,4,5]. The topological space is a set of points/atoms with the corresponding neighborhood for each point, of which axiomatic existence is accepted [6,7,8]. All biological variables, reflecting either a discrete or a continuous presence and interaction of biological entities, can be represented and measured in Euclidian space [9]. Topological spaces can be described, up to homeomorphism, by their topological properties, which are invariant under homeomorphism. A topological space that is locally Euclidian is called a manifold. As the Euclidian space, a manifold can have n-dimensions [10]. The notion of a manifold is pivotal to many parts of geometry because it allows complicated structures to be described in terms of well-understood topological properties of elementary spaces. Manifolds naturally arise as solution sets of systems of equations and as graphs of functions. Manifold spaces may be weighted to “carry” some properties, such as bioactivity [11]. Particularly, receptor-ligand interaction carries a vast range of properties. Such properties can be ligand-receptor interaction energies, ligand conformational energies, and lipophilic-related properties, respectively. Those interactions can be described as stated before by manifolds [12,13]. Thus, a variable describing the bioactivity can be represented using manifolds (i.e., by computing a set of equations that describe or are related to the variable) [14]. Pharmacological bioactivity results from the biodynamics and biokinetics of a particular drug; biodynamics characterizes the interaction between the drug and its target, and biokinetics describes the path of the drug to its target; thus, these two properties contain all the systems involved in the drug-receptor interactions. Overall, these activities can be considered to form the drug space [15,16]. There is a vast number of distinct molecules. However, not all molecules have biological activity. So, the drug space is a specific region that is only a part of a vast region of the chemical space [17,18,19]. Chemical space is a conceptualization in cheminformatics that refers to the space including all possible molecules and chemical compounds defined by a given set of construction rules and boundary conditions. Therefore, all possible compounds (known and theoretical) and chemical elements are components of chemical space [20,21,22]. Theoretical chemical space in cheminformatics refers to the space of potentially pharmacologically active molecules with a size order of 10^60^ molecules also known as chemical reaction space [23,24]. Concrete chemical space envelopes, as registered by Chemical Abstract Service in July 2009, 49.037.297 organic and inorganic molecules, and reported on November 11th, 2013, over 75 million structures were cited in the journals. Exploration of chemical space can be performed computationally—in silica or experimentally—by chemical reactions. Both processes in silica data mining or database search and chemical reaction, respectively, return new compounds that finally lead to chemical space exploration. Exploration of the chemical space is performed by using in silica databases of virtual molecules, visualized by projecting the multidimensional molecular property space in lower dimensions, characterized by physicochemical measurable/quantifiable properties. As stated before, the generation of valid chemical structures using any means (experimental or in silica) leads to novel compounds [25,26,27,28]. Moving through the chemical space is performed by generating stoichiometric combinations of 12 electrons (all stoichiometric inorganic materials) and atomic nuclei to approach all possible topology isomers to give construction rules. In cheminformatics, such techniques include structure generation engines, fragmentation engines, and more discrete techniques such as conformational searching and analyzing techniques such as molecular dynamics and docking [29,30].

In the experimental situational world, physicochemical properties are the projection of chemical properties in the known chemical space; being only projections, these properties are often not unique (degenerate, i.e., different molecules with the same mass) thus different molecules can exhibit similar properties [31]. Material design and drug discovery both involve the exploration of chemical space.

Infectious diseases significantly cause morbidity and mortality, which the current antimicrobial resistance crisis can worsen [32]. Antimicrobial resistance is evolving rapidly. In addition, antimicrobials are valuable resources that enhance the prevention and treatment of infections. As resistance diminishes this resource, it is a societal goal to minimize resistance and reduce forces that produce resistance. Genetic recombination allows bacteria to rapidly disseminate genes encoding for antimicrobial resistance within and across species [33]. Antimicrobial use creates a selective evolutionary pressure, which leads to further resistance. Antimicrobial stewardship, best use, and infection prevention are the most effective ways to slow the spread and development of antimicrobial resistance [34,35]. Thus, developing new drugs is crucial in managing antibiotic resistance [36,37].

Being a bioactive and relatively easy-to-derivate molecule and taking into account previous published research, the chemical space of benzoquinolines was chosen for further research. In this work, the chemical space of benzoquinolines is explored. Topological space, chemical space, and bioactivity space are discussed.

## 2. Results

In a previous research study, we presented some benzoquinoline salts derivatives’ synthesis and antimicrobial (antibacterial and antifungal) activity against *S. aureus*, *E. coli*, and *C. albicans* (Figure 1).

Some of the obtained salts, namely **3i**, **3j**, and **3n**, have potent antimicrobial activity on S. aureus, E. coli, and C. albicans strains. From the obtained molecular docking studies (present work), we noticed that salts **3i**, **3j**, and **3n** have the best fit in complex with the microbial ATP synthase. In the binding pocket of bacterial ATP synthase, a powerful hydrogen bond between the oxygen atom (from the CO group) and aminoacid ARG-364 is formed, stabilizing the complex salt BQS (benzoquinolines) **3j**—ATP synthase. From the docking studies using ligand interaction viewing dialog, we also noticed that ARG 357 shares electrons with heterocycle aromatic moiety [38].

In Table 1, the single pharmacophore hypothesis is generated using cartesian coordinates, representing the type of **3j** atoms.

The pharmacophore data from Table 1 reveal the importance of the acceptor group and the positive ionic group surrounded by aromatic centers.

Using molecular descriptors and ligand–receptor docking data (regarding ATP synthase), QSAR models were developed. The five best-fit QSAR models are shown in Table 2, Table 3 and Table 4 represent QSAR models for bioactivities on *S. aureus*, *E. coli*, and *C. albicans*.

A multiple linear regression (MLR) QSAR model for *S. aureus* PDB ID AF_AFP99112F1 [39] (targeting ATP synthase) was built using the five descriptors set (steric energy—steric energy of the compound kcal/mol; H bond—number of hydrogen bounds in the compound; heavy atoms—number of heavy atoms; N—number of nitrogen atoms, UnH bond—unbonded hydrogen atoms; see Table 2) and shows a Pearson correlation (r) = 0.972, Pearson correlation squared (r^2^) = 0.945704, Spearman rank correlation (*p*) = 0.964286, mean squared deviation (MSD) = 1.38611, root mean square deviation (RMSD) = 1.17733, and cross-validated square (q^2^) = 0.945704, respectively. The MLR equation is y = 0.9457x + 0.8995 (Figure 1). After performing the principal component analysis, it was observed that the QSAR model shows that nitrogen atoms and unidirectional hydrogen bonds have a crucial role in the activity of *S. aureus*, explaining 69% of the phenomena (antimicrobial activity). A heavy atom’s presence significantly increases the bioactivity of the compound (r^2^ = 0.90). Steric and hydrogen bond energy can explain less than 5% of phenomena (antimicrobial activity). Ligand composition is imperative for activity on *S. aureus*, while ligand conformation and bonding do not have such a significant impact.

*E. coli* was selected while previous studies show that BQS shows good activity against it. A multiple linear regression (MLR) QSAR model for *E. coli* targeting microbial *E. coli* ATP synthase PDB ID 6OUQ [40] built using five descriptors (see Table 3) shows a Pearson correlation (r) = 0.947, Pearson correlation squared (r^2^) = 0.896, Spearman rank correlation (*p*) = 0.896, mean squared deviation (MSD) = 2.039, root mean square deviation (RMSD) = 1.428, and cross-validated square (q^2^) = 0.896, respectively. The MLR equation is y = 0.8968x + 1.3927 (Figure 1). The QSAR model suggests that the ligand structure has a critical role in the activity of *E. coli* (Figure 1). PCA showed that the degrees of freedom (DOF) and N atoms can explain 70% of BQS3 bioactivity in *E. coli*. In addition, the torsional energy of the ligand halogen atom and unidirectional hydrogen bonds has been implied.

A multiple linear regression (MLR) QSAR model for *C. albicans* (selected due to its study in previous work) targeting ATP synthase was built using five descriptors (Table 4) shows a Pearson correlation (r) = 0.9867, Pearson correlation squared (r^2^) = 0.9737, Spearman rank correlation (*p*) = 0.964, mean squared deviation (MSD) = 0.548, root mean square deviation (RMSD) = 0.740, and cross-validated square (q^2^) = 0.975, respectively. The MLR equation is y = 0.973x + 0.4678 (Figure 1).

By using docking energies or the rest of the molecular descriptors, a modest QASAR model has been obtained (r^2^ < 0.6). The activity on *C. albicans* can be explained in 76% of cases using a composed molecular descriptor: partial charges and van der Waals surface (PEOE_VSA + 2). The phenomena are explained by combined descriptors that correlate with partial charges, molecular surface, and volume.

In Figure 1, scatter plots represent the observed and predicted activities on *S. aureus*, *E. coli*, and *C. albicans*. Observed and predicted activities are roughly following each other. One point is an outlier for *S. aureus* and *E. coli* due to lack of activity.

Virtual screening results using hypothesis A1H2P3R4R5R6R7 are represented in Figure 2 (the first ten best-fit compounds). The screening retrieved more than 7000 compounds that satisfy at least four criteria of the hypothesis A1P2R3R4R5R6R7.

Screening results retrieved many heterocyclic compounds with nitrogen. As stated before, the screening template used was the pharmacophore hypothesis developed with the help of the QSAR model. These findings follow the computed QSAR model and suggest exploring a distinct class of compounds with superior activity on discussed strains.

Salts in interaction with bacterial ATP synthase screen cluster results are represented in Figure 3.

The single pharmacophore hypothesis of salts in complex with TOPO II is represented in Table 5; the pharmacophore hypothesis is generated using cartesian coordinates, and the type of **3j** atoms is represented.

The pharmacophore data from Table 5 reveal the importance of the acceptor group (i.e., C=O, C-O-C, C-NR, C=NR, C-OH, C-NH, C=NH) and the positive ionic group (i.e., Al^3+^, Zn^2+^, NH^4+^,) surrounded by aromatic centers. Table 6, Table 7 and Table 8 represent QSAR models for bioactivities considering TOPO II as a target on *S. aureus*, *E. coli*, and *C. albicans*.

A multiple linear regression (MLR) QSAR model (targeting bacterial ATP synthase) for *S. aureus* was built using five descriptors (see Table 6) and shows a Pearson correlation (r) = 0.960, Pearson correlation squared (r^2^) = 0.921788, Spearman rank correlation (*p*) = 0.88, mean squared deviation (MSD) = 1.996, root mean square deviation (RMSD) = 1.413, and cross-validated square (q^2^) = 0.921788, respectively. The MLR equation is y = 0.9218x + 1.2957 (Figure 4).

PCA showed that the presence of an N atom can explain 42% of the phenomena. The degrees of freedom also have a crucial role in bioactivity. Pose energy and torsion energy count for the rest of the phenomena.

A multiple linear regression (MLR) QSAR (targeting bacterial ATP synthase) model for *E. coli* built using the five descriptors (see Table 7) shows a Pearson correlation (r) = 0.92798, Pearson correlation squared (r^2^) = 0.882256, Spearman rank correlation (*p*) = 0.91785, mean squared deviation (MSD) = 2.32741, root mean square deviation (RMSD) = 1.52559, and cross-validated square (q^2^) = 0.882256, respectively. The MLR equation is y = 0.8823x + 1.5895 (Figure 4).

The QSAR model suggests that the activity of *E. coli* of **BQS 3** in a complex with bacterial TOPO II depends on the number of degrees of freedom (DOF), the number of nitrogen (N) atoms, and the torsional and total energy of the complex.

A multiple linear regression (MLR) QSAR model for *C. albicans* built using five descriptors (Table 8) shows a Pearson correlation (r) = 0.9867, Pearson correlation squared (r^2^) = 0.9737, Spearman rank correlation (*p*) = 0.91785, mean squared deviation (MSD) = 2.32741, root mean square deviation (RMSD) = 1.52559, and cross-validated square (q^2^) = 0.882256, respectively. The MLR equation is y = 0.9737x + 1.5895 (Figure 4).

The QSAR model shows that bioactivity on *C. albicans* of **BQS** salts **three** could be explained by the membrane permeability properties of the compounds (characterized by using membrane permeability descriptors: see Table 8). In addition, findings suggest that the polarizability of the ligand has a crucial role.

Figure 4 shows scatter plots of observed and predicted activities on *S. aureus*, *E. coli*, and *C. albicans*.

Observed and predicted antibacterial activities follow each other. An outlier is observed in the case of *S. aureus* and *E. coli* due to a lack of activity. Overall, the scatter plot demonstrates a correct prediction of the variable (bioactivity—antibacterial activity) by using the multiple linear regression (MLR) QSAR models.

Virtual screening results using hypothesis A1P3R4R5R6R7 are represented in Figure 5 (first ten best-fit compounds). The screening retrieved more than 9000 compounds that satisfy at least four criteria of the hypothesis A1P2R3R4R5R6R7.

Screening results show that compounds with heterocycles in the structure and halogen atoms bound with aromatic rings satisfy the hypothesis A1P2R3R4R5R6R7 and can be active on bacterial ATP synthase and topoisomerase II. Both enzymes are crucial in bacterial growth and multiplication and are preferable antibiotic targets.

Salts in interaction with TOPO II screen cluster results are represented in Figure 6.

## 3. Discussion

Regarding screening compounds’ structural properties, it is noted that both series of the compound screen against the two bacterial enzymes, TOPO II and ATPase, have the same structural properties. For example, the number of heavy and hydrogen and carbon atoms is similar (Figure 7).

Chemical or molecular similarity refers to the similarity of chemical elements, molecules, or chemical compounds concerning functional qualities. Biological effects are quantified using the biological activity of a compound. The function can be related to the chemical activity of a compound. The chemical similarity is often described as an inverse of a distance measure in descriptor space (i.e., for inverse distance measures are the molecule kernels) [41]. The concept is essential in cheminformatics [42]. It plays a vital role in approaches to predicting the properties of chemical compounds, designing molecules with predefined properties, and, especially, conducting drug design studies by screening large databases containing structures of available (or potentially available) chemicals. These concepts are based on the similar property principle of Johnson and Maggiora, which states that similar compounds have similar properties [43]. The similarity-based [44] virtual screening states that all compounds in a database similar to a query compound have similar biological activity. However, this hypothesis is not always valid, while the set of retrieved compounds is often considerably enriched with actives [45,46]. Molecular structures are usually represented by structural keys or fixed or variable-size molecular fingerprints. Molecular screens and fingerprints can contain various two-dimensional or three-dimensional information.

Chemical similarity can be expanded to similarity network theory in intense, extensive library virtual screening clusters of compounds (Figure 8). The theory state that descriptive network properties and graph theory can be applied to analyze sample chemical space, estimate chemical diversity, and predict drug target. In addition, 3D chemical similarity networks based on 3D ligand conformation have been developed, which can be used to identify scaffold-hopping ligands. Cluster analysis resulting after docking screening against bacterial TOPO II and ATPase retrieves one cluster for both enzymes with relatively no outliers (Figure 8). These findings suggest a common chemical space for both bioactivities.

## 4. Materials and Methods

Using two essential compounds synthesized and tested [47], two single pharmacophore hypotheses were developed for best salts compounds with bacterial topoisomerase II (TOPO II) and bacterial adenosine triphosphate synthase (ATP-synthase) as targets. The crystal structures used for docking studies were 1AB4 for TOPO II [48] and 1EF0 for ATP synthase [49] In addition, a single pharmacophore hypothesis was developed using the Schrodinger software package [50].

An extensive virtual screening (over 2 million compounds) was performed using the resulting hypothesis against a diverse molecular data set. The data set used in the screening was retrieved from ChEMBL Database [51]. Library compounds were checked for errors, charges were corrected, energy was minimized, and H atoms were properly attributed using the LigPrep software package. The number of hypotheses hits was set to 4 of 7 (hypothesis with 7 crucial domains/functional groups). Molecular descriptors and drug-like property filters were computed for screening results. Using descriptors data, the chemical space was characterized. Descriptors were computed using Schrodinger and MOE software [52,53]. Finally, multidimensional clusters of the screening set were computed. Descriptors used in representing the clusters were as follow: phase group fitness, phase screen score, phase fitness, phase volume score, phase vector score, phase align score, phase conformational index, phase f3d relative energy, and a series of molecular descriptors: molecular weight, VSA in function with polarizability, hydrofily, van der walls volume, vdw_vol, vdw_area, vista, VdistEq, VAAdjEq, TPSA, SMRVA, SMR, SlogP, number of rings, PEOE VSA, oprviolation, opera ring, opera lead-like molecular refractivity, log P(o/w), Kier index, BCUT.

Docking studies were performed using screening data sets for TOPO II and ATP synthase. PDB structures of TOPO II and ATP synthase were used. Binding sites used were retrieved from the literature [54,55] and using the online docking server SWISS Target [56], receptor structures were retrieved from the PDB: 6WLZ for ATP synthase [16] and 3KSB for TOPO II [57]. The PDB structures were energetically minimized, charges corrected, names corrected, and potential energy recomputed. Cofactors and Aa chains were kept, and ligands and water molecules were removed. Screening resulted from compounds introduced as SDF structures [58,59]. The maximum number of cavities detected was 5, corresponding to molecular theory’s binding site. The cavity with the most significant volume (Å^3^) was chosen for each target.

MOE 2009 software and its methodology were used in the docking procedure [60,61,62]. Cartesian coordinates for the two binding sites are as follows: ATP synthase: x45.94 Å, y46.91 Å, z198.20 Å; TOPO II: x-16.33 Å, y43.11 Å, z-34.50 Å. For computational and action mechanism reasons, only the protein corona of ATP synthase was used in building the in silico system. Docking was validated by redocking the ligands present in PDB target structures. For this step, ligands were cut from the crystallographic obtained structures, converted to SDF files, minimized, and charges corrected. Ligands were then docked onto the free ligand PDB structures. Compounds were ordered using the docking score. The best ten compounds were presented and discussed for each molecular target ( TOPO II and ATP-synthase, respectively). Multidimensional clusters using docking energies were computed and represented as scatter plots. Docking-derived data used for computing the clusters were as follows: the number of ligands hybridizing Csp2 atoms, the number of ligands hybridizing Csp3 atoms, internal ligand cofactor energy, internal ligand water energy, total complex energy, steric energy, torsional ligand energy, electron energy, hydrogen bond energy, total ligand energy, ligand steric energy, ligand van der walls energy. All energies were expressed in kcal/mol.

Chemical space is represented using a multidimensional cluster representation. The clusters were generated using molecular descriptors data, and docking energies resulted from the docking studies of compounds.

## 5. Conclusions

The benzoquinoline chemical space is characterized by the lipophilicity of the molecules and the number of heteroatoms. In addition, aromatic rings play a crucial role in defining their property space.

Screening results show that compounds with heterocycles in structure and halogen atoms bound with aromatic rings satisfy the hypothesis A1P2R3R4R5R6R7 and can be active on ATP synthase and topoisomerase II. Furthermore, antimicrobial and antitumoral activity seems to be related to the lipophilic properties of the compounds. Additionally, the geometry of the compound is essential in molecule bioactivity. Lastly, as a result of screening, these properties seem to be concentrated in one cluster rather than dispersed along various clusters.

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
