# Peer review of "Benzoquinoline Chemical Space: A Helpful Approach in Antibacterial and Anticancer Drug Design"

_molecules, 2023, doi:10.3390/molecules28031069_

Round 1

Reviewer 1 Report

General comments:

I)               The abstract was too general and did not indicate clearly the aims and the main findings.

II)             The introduction did not justify the research performed. It was unclear why benzoquinolines were selected and which problems remain to be solved.

III)            Several sentences lacked justification and references.

IV)           There was insufficient information what was the target of benzoquinolines. It  was unclear if benzoquinolines shall target microbial or mammalian enzymes. There were no explanations how microbial activities (and which ones) could be assessed. 

V)             The overall manuscript was too broad and did not add significant information, as reflected in the discussion and in the conclusion. 

Minor comments:

1)    Abstract, p1: Delete “Benzoquinolines are an essential backbone used in medicinal chemistry to design novel  molecules with potential antibacterial and antitumoral properties. s. However, molecules based on  this molecular typology failed in Phase I trials.”

2)    Abstract, p1: Avoid general statements. Delineate exactly the research aims. Specify drug targets and Rephrase “This computational study emphasizes benzoquinone drug space.”

3)    Abstract, p1: The sentence was unclear. Replace “virtual” by another word. Specify topological, chemical and bioactivity spaces to support “Topological, chemical, and bioactivity spaces are explored using virtual methodologies  based on virtual screening and scaffold hopping.”

4)    Abstract, p1: add more information to support “Results show a poly-morphological chemical space 16 that suggests distinct characteristics.”

5)    Abstract, p1: Replace “novel” by chemical names, and identify what are the drug targets. “Lastly, novel chemical compounds with drug-like potential have been identified.”

6)    Introduction, p1: add references to support “Bioactivity, or biological activity, outlines the property of a stimulus (radiation, molecule, biomolecule, drug) to evoke a biological response from living matter.”

7)    Introduction, p1: add references to support “In pharmacology, this activity is determined by ligand molecule pharmacophores, constituting and shaping the interactions between the ligand molecule and its target (i.e., receptor).”

8)    Introduction, p1: Add references to support “Topological spaces can be described, up to homeomorphism, by their topological properties, which are invariant under homeomorphism.”

9)    Introduction, p1: Add explanations and references to support “Manifold spaces may be weighted to,, carry" some properties, such as bioactivity.”

10) Introduction, p1: Add explanations and references to support “Thus a variable describing the bioactivity can be represented using manifolds.”

11) Introduction, p1: Avoid general statements, add specific information and references to support “Pharmacological bioactivity results from the biodynamics and 35 biokinetics of a particular drug; these activities can be considered to form the drug space.”

12) Introduction, p1: Justify and add references to support “Drug space is a particular region that is only a part of a vast region of the chemical space.”

13) Introduction, p1: Add explanations and references to “Chemical space is a conceptualization in cheminformatics that refers to the space including all possible molecules and chemical compounds defined by a given set of construction rules and boundary conditions.”

14) Introduction, p1: It was unclear what is the justification of 1060 molecules and how the number was selected. Add explanations and references to support “Theoretical chemical space in cheminformatics refers to the space of potentially pharmacologically active molecules with a size order of 1060 molecules.”

15) Introduction, p1: Justify and add references to support “Exploration of chemical space can be performed computationally - in silica or experimentally – by chemical reactions.”

16) Introduction, p2: Justify “Exploration of chemical space is performed by using in silica database of virtual molecules, visualized by projecting the multidimensional molecular property space in lower dimensions, characterized by physicochemical measurable/quantifiable properties [4].”

17) Introduction, p2: It was unclear why 12 electrons were selected. Justify and add references to support “Moving through the chemical space is performed by generating stoichiometric combinations of 12 electrons and atomic nuclei to approach all possible topology isomers to give onstruction rules.”

18) Introduction, p2: The sentence was too broad and unclear. Rephrase “In the real world, physicochemical properties are the projection of chemical properties in the known chemical space; being only projections, these properties are often not unique (i.e., degenerate), and different molecules can exhibit similar properties [6].”

19) Introduction, p2: The sentence does not bring useful information. Delete “Materials design and drug discovery both involve the exploration of chemical space.”

20) Introduction, p2: It was unclear why benzoquinolines were selected and what are the problems to justify the investigation. Justify “In this work chemical space of benzoquinolines is explored. Topological space, chem- 58 ical space, and bioactivity space are discussed.”

21) Results, p2: Specify which type of antimicrobial activity to support “Some of the obtained salts, namely3i, 3j, and 3n, have potent antimicrobial activity.”

22) Results, p2: Specify the specie origin of ATP synthase (microbial or mammalian ?)  and add references to support “From the obtained molecular docking studies, we noticed that salts 3i, 3j, and 3n have the best fit in complex with ATP synthase.”

23) Results, p2: Add explanations to support “We also noticed that ARG 357 shares electrons with heterocycle aromatic moiety [7].”

24) Results, p2: Specify BQS in “In the binding pocket of ATP synthase, a powerful hy- drogen bond between the oxygen atom (from CO group) and aminoacid ARG-364 is formed, stabilizing the complex salt BQS 3j - ATP synthase”

25) Results, Table 1: Specify abbreviations, A1,P2,R3,R4,R5,R6,R7 to support Table 1.

26) Results, add comments to support findings from Table 2, 3, and 4 in “Tables 2, 3, and 4 represent 78 QSAR models for bioactivities on S. aureus, E. coli, and C. Albicans.”

27) Results, p3: Explain what are the 5 descriptors,  why S. aureus was selected, it was unclear if the target enzyme was the ATP synthase to support “Multiple linear regression (MLR) QSAR model for S. aureus was build using the 5 descriptors set (see Table 2) and shows a Pearson correlation (r)= 0.972, Pearson correlation squared (r2)= 0.945704, Spearman rank correlation (p)= 0.964286, Mean squared devi- 84 ation (MSD)= 1.38611, Root mean square deviation (RMSD)= 1.17733 and Cross validated 85 square (q2)= 0.945704 respectively. The MLR equation is y= 0.9457x+0.8995 (Figure 1).”

28) Results, p3: It was unclear how the percentage of the phenomena was computed and to which phenomena it referred. Justify, add explanations to support  “The  QSAR model shows that nitrogen atoms and unidirectional hydrogen bonds have a crucial role in the activity of S. aureus, explaining 69% of the phenomena. A heavy atom's presence significantly increases the bioactivity of the compound (r2= 0.90). Steric and hydrogen bond energy can explain less than 5 % of phenomena.”

29) Results, p3: The sentence was unclear. It was unclear what are the ligands to support “Ligand composition is imperative inactivity on S. aureus, while ligand conformation and bonding do not significantly impact.”

30) Results, p3: Indicate why E.Coli was selected and if the enzyme target was the ATP synthase to support “ultiple linear regression (MLR) QSAR model for E. coli build using 5 descriptors (see Table 3) shows a Pearson correlation (r)= 0.947, Pearson correlation squared (r2)= 0.896, Spearman rank correlation (p)= 0.896, Mean squared deviation (MSD)= 2.039, Root mean square deviation (RMSD)= 1.428 and Cross validated square (q2)= 0.896 respectively. The MLR equation is y= 0.8968x+1.3927 (Figure 1).”

31) Results, p3: Specify the ligand to support “The QSAR model suggests that the ligand structure has a critical role in the activity of E. coli.”

32) Results, p3: Add explanations to support “The degrees of freedom (DOF) 100 and N atoms can explain 70% of BQS3 bioactivity in E. coli. Also, the torsional energy of 101 the ligand halogen atom and unidirectional hydrogen bonds has been implied.”

33) Results, p4: Indicate why C. Albicans was selected and if the enzyme target was the ATP synthase to support “Multiple linear regression (MLR) QSAR model for C. Albicans build using 5 descriptors (Table 4) shows a Pearson correlation (r)= 0.9867, Pearson correlation squared  (r2)= 0.9737, Spearman rank correlation (p)= 0.964, Mean squared deviation (MSD)= 0.548, Root mean square deviation (RMSD)= 0.740 and Cross validated square (q2)= 0.975 respectively. The MLR equation is y= 0.973x+0.4678 (Figure 1).”

34) Results, Fig 1, p4: Indicate in the Figure-1 legend what are the observed and predicted values to support “Figure 1. Scatter plot of observed (on ox axis) and predicted (on oy axis) MIC for: a. S. aureus; b. E. 115 coli, and c. C. albicans.”

35) Results, p4: It was unclear which activities and how predicted activities can be obtained to support “In Figure 1, scatter plots represent the observed and predicted activities on S. aureus, E. coli, and C. Albicans. Observed and predicted activities are roughly following each other.  One point is an outlier for S. aureus and E. coli due to lack of activity.”

36) Results, p4: It was unclear from which data base were retrieved the 7000 compounds and if they were all benzoquinolines to support “The screening retrieved more than 7000 compounds that satisfy at least four criteria of the hypothesis A1P2R3R4R5R6R7; see supplemental files.”

37) Results, p5: The sentence was too broad. Specify types of activity and Justify “Screening results retrieved many heterocyclic compounds with nitrogen. These findings follow the computed QSAR model and suggest exploring a distinct class of compounds with superior activity on discussed strains.”

38) Results, Figure 2, p6: Indicate the specie origin of the ATP synthase to support the Figure-2 legend in “Virtual screening cluster of compounds for BQS salts in interaction with ATP synthase is represented as a scatter plot: x- and y-axis represent coordinates of each compound characterized by its properties: relative energy, Phase vector score, phase volume score (see supplemental files).”

39) Results, p6: Specify the acceptor and the positive ionic groups to support “The pharmacophore data from Table 6 reveal the importance of the acceptor group 142 and the positive ionic group surrounded by aromatic centers.”

40) Results, p6: Specify activities in “Tables 7, 8, and 9 represent QSAR models for bioactivities on S. aureus, E. coli, and C. Albicans. ”

41) Results, p7: It was unclear to which type of ATP synthase the data from table 7 refers to in “Multiple linear regression (MLR) QSAR model for S. aureus was build using 5 descriptors (see Table 7) and shows a Pearson correlation (r)= 0.960, Pearson correlation squared (r2)= 0.921788, Spearman rank correlation (p)= 0.88, Mean squared deviation 149 (MSD)= 1.996, Root mean square deviation (RMSD)= 1.413 and Cross validated square 150 (q2)= 0.921788 respectively.”

42) Results, p7: It was unclear which type of bioactivities and which phenomena to support“QSAR models show that the presence of an N atom can explain 42% of the phenomena. The degrees of freedom also have a crucial role in bioactivity. Pose energy and torsion energy count for the rest of the phenomena. 

43) Results, p7: It was unclear to which type of ATP synthase the data from table 8 refers to in “Multiple linear regression (MLR) QSAR model for E. coli build using the 5 descriptors a Pearson correlation (r)= 0.92798, Pearson correlation squared (r2)= 0.882256, Spearman rank correlation (p)= 0.91785, Mean squared deviation (MSD)= 2.32741, Root mean square deviation (RMSD)= 1.52559 and Cross validated square (q2)= 0.882256 respectively. The MLR equation is y= 0.8823x+1.5895 (Figure 4).”

44) Results, p7: Specify TOPO II in “The QSAR model suggests that the activity of E. coli of BQS 3 in a complex with TOPO II depends on the number of degrees of freedom (DOF), the number of nitrogen (N) atoms, and the torsional and total energy of the complex.”

45) Results, p7: It was unclear to which type of ATP synthase the data from table 9 refers to in “Multiple linear regression (MLR) QSAR model for C. albicans build using 5 descriptors (Table 9) shows a Pearson correlation (r)= 0.9867, Pearson correlation squared  (r2)= 0.9737, Spearman rank correlation (p)= 0.91785, Mean squared deviation (MSD)= 2.32741, Root mean square deviation (RMSD)= 1.52559 and Cross validated square (q2)=  0.882256 respectively. The MLR equation is y= 0.9737x+1.5895 (Figure 4).”

46) Results, p7: It was unclear how membrane permeabilities were assessed to support “QSAR model shows that bioactivity on C. Albicans of BQS salts three could be explained by the membrane permeability properties of the compounds. In addition, findings suggest that the polarizability of the ligand has a crucial role.”

47) Results, Fig. 4, p8: Specify the observed and predicted values in the Figure-4 legend to support  “Figure 4. Scatter plot of observed (on ox axis) and predicted (on oy axis) MIC for a. S. aureus; b. E. 177 coli, and c. C. albicans.”

48) Results, p8: Specify which type of bioactivity to support “Overall the scatter plot demonstrates a correct prediction of the variable (bioactivity) by using the multiple linear regression 181 (MLR) QSAR models.”

49) Results, p8: It was unclear from which data base was selected the 9000 compounds and why the previous findings were based from 7000 compounds to justify “The screening retrieved more than 9000 compounds that satisfy at least four criteria of the hypothesis A1P2R3R4R5R6R7; see supplemental files.”

50) Results, p8: Explain why topoisomerase II was selected and Indicate the specie origin of ATP synthase and Topoisomerase II. To support “Screening results show that compounds with heterocycles in structure and halogen 188 atoms bound with aromatic rings satisfy the hypothesis A1P2R3R4R5R6R7 and can be 189 active on ATP synthase and Topoisomerase II.”

51) Discussion, p9: Indicate the specie origin of ATP synthase, and TOPO II. “Regarding screening compounds' structural properties, it is noted that both series of the compound screen against the two enemies, TOPO II and ATPase, have the same structural properties.”

52) Discussion, Fig 7, p10: Indicate the specie origin of ATP synthase, and TOPO II to support Figure-7 legend in . “Structural composition of molecular dataset resulted after screening”

53) Materials and Methods, p11: Indicate the species of TOPO II and ATP-synthase“Using to support, indicate chemical structures of the two compounds to support  “Two essential compounds synthesized and tested [8], two single pharmacophore hypotheses were developed for best salts compounds with topoisomerase II (TOPO II ) and Adenosine triphosphate synthase (ATP-synthase) as targets.”

54) Materials and Methods, p11: It was unclear how the screening was performed and what was the target to support “An extensive virtual screening ( over 2 million compounds ) was performed using the resulting hypothesis against a diverse molecular data set. The data set used in the screening was retrieved from ChEMBL Database [10].”

55) Materials and Methods, p11: Indicate the specie of the enzymes in “TOPO II and ATP -synthase. PDB structures of TOPO II and ATP synthase were used.”

56) Conclusions, p12: It was unclear if compounds shall target the microbial enzymes or the human enzymes since there were insufficient information how biological activities were measerd. Rephrase, add information to support  “Screening results show that compounds with heterocycles in structure and halogen  atoms bound with aromatic rings satisfy the hypothesis A1P2R3R4R5R6R7 and can be active on ATP synthase and Topoisomerase II. Furthermore, antimicrobial and antitumoral activity seems to be related to the lipophilic properties of the compounds.”

Author Response

The abstract was too general and did not indicate clearly the aims and the main findings The introduction did not justify the research performed. It was unclear why benzoquinolines were selected and which problems remain to be solved. Several sentences lacked justification and references. There was insufficient information what was the target of benzoquinolines. It  was unclear if benzoquinolines shall target microbial or mammalian enzymes. There were no explanations how microbial activities (and which ones) could be assessed. The overall manuscript was too broad and did not add significant information, as reflected in the discussion and in the conclusion. 

Comment : 1   Abstract, p1: Delete “Benzoquinolines are an essential backbone used in medicinal chemistry to design novel  molecules with potential antibacterial and antitumoral properties. s. However, molecules based on  this molecular typology failed in Phase I trials.”

Response: corrected. The following sentence was added: Benzoquinolines are used in many drug design projects as starting molecules which are subject to derivatization. 

Comment: 2)    Abstract, p1: Avoid general statements. Delineate exactly the research aims. Specify drug targets and Rephrase “This computational study emphasizes benzoquinone drug space.”

Response 2: corrected as suggested> The following phrases were added: This computational study aims to characterize e benzoquinone drug space to ease future drug design processes based on these molecules.

Comment:3)    Abstract, p1: The sentence was unclear. Replace “virtual” with another word. Specify topological, chemical and bioactivity spaces to support “Topological, chemical, and bioactivity spaces are explored using virtual methodologies  based on virtual screening and scaffold hopping.”

Response 3: corrected as suggested. Virtual was replaced with computational.

Comment:4)    Abstract, p1: add more information to support “Results show a poly-morphological chemical space 16 that suggests distinct characteristics.”

Response 4: The following phrases were added: The chemical space seems to be correlated with properties like steric energy, the number of hydrogen bonds, the presence of halogen atoms, and membrane permeability-related properties. 

Comment: 5)    Abstract, p1: Replace “novel” by chemical names, and identify what are the drug targets. “Lastly, novel chemical compounds with drug-like potential have been identified.”

Response 5 :  correted . The following line was added: Lastly, novel chemical compounds with drug-like potential have been identified, like oxadiazole methybenzamide and floro methylcyclohexane diene. 

Comment: 6)    Introduction, p1: add references to support “Bioactivity, or biological activity, outlines the property of a stimulus (radiation, molecule, biomolecule, drug) to evoke a biological response from living matter.”

Response 6: the following references were added: 1. M Florencia Sánchez, Sylvia Els-Heindl, Annette G Beck-Sickinger, Ralph Wieneke, Robert Tampé. Photoinduced receptor confinement drives ligand-independent GPCR signaling. Science. 2021; 26;371(6536)

  1. Eiki Kanbe , Akihiro Abe, Masayuki Towatari, Tsutomu Kawabe, Hidehiko Saito, Nobuhiko Emi. DR1-like element in human topoisomerase IIalpha gene involved in enhancement of etoposide-induced apoptosis by PPARgamma ligand. Exp Hematol.2003; 31(4):300-8.

Comment: 7)    Introduction, p1: add references to support “In pharmacology, this activity is determined by ligand molecule pharmacophores, constituting and shaping the interactions between the ligand molecule and its target (i.e., receptor).”

Response 7: The following references were added:

  1. L N Harris 1, L Yang, C Tang, D Yang, R Lupu. Induction of sensitivity to doxorubicin and etoposide by transfection of MCF-7 breast cancer cells with heregulin beta-2. Clin Cancer Res. 1998;4(4):1005-12.
  2. Wojciech Bocian , Beata Naumczuk , Magdalena Urbanowicz , Jerzy Sitkowski , Elżbieta Bednarek , Katarzyna Wiktorska , Anna Pogorzelska , Ewelina Wielgus , Lech Kozerski. Insight on the Interaction between the Camptothecin Derivative and DNA Oligomer Mimicking the Target of Topo I Inhibitors. Molecules. 2022;;27(20):6946.
  3. Tahmeena Khan, Rumana Ahmad , Iqbal Azad , Saman Raza , Seema Joshi, Abdul Rahman Khan. Computer-aided drug design and virtual screening of targeted combinatorial libraries of mixed-ligand transition metal complexes of 2-butanone thiosemicarbazone. Comput Biol Chem. 2018; 75:178-195.

Comment: 8)    Introduction, p1: Add references to support “Topological spaces can be described, up to homeomorphism, by their topological properties, which are invariant under homeomorphism.”

Response8: the following references were added :

  1. Beata Szefler, Mircea V DiudeaSpongy Nanostructures. J Nanosci nanotechnology. 2017; 17(1):323-28.
  2. Claudiu N Lungu, Mircea V Diudea , Mihai V Putz, Ireneusz P Grudziński. Linear and Branched PEIs (Polyethyl-enimines) and Their Property Space. Int J Mol Sci . 2016; 13;17(4):555.

Comment: 9)    Introduction, p1: Add explanations and references to support “Manifold spaces may be weighted to,, carry" some properties, such as bioactivity.”

Response 9: the following line was added, and their references: Particularly receptor ligandin interaction carries a vast range of properties. Those interactions can be described as stated before by manifllds[12,13]. 

  1. Ken J Ishii, Cevayir Coban, Shizuo Akira. Manifold mechanisms of Toll-like receptor-ligand recognition. J Clinical Imunolo-gy. 2005;25(6):511-21.
  2. Andras Piffko , Christian Uhl , Peter Vajkoczy , Marcus Czabanka , Thomas Broggini. EphrinB2-EphB4 Signaling in Neuroon-cological Disease. Int J Mol Sci. 2022 Jan 31;23(3):1679.

Comment: 10) Introduction, p1: Add explanations and references to support “Thus a variable describing the bioactivity can be represented using manifolds.”

Response 10: the following reference was added

  1. Vince M Lombardo , Christopher D Thomas, Karl A Scheidt. A tandem isomerization/prints strategy: iridium(III)/Brønsted acid cooperative catalysis. Angew Chem Int Ed 2013;52(49):12910-4.

Comment: 11) Introduction, p1: Avoid general statements, add specific information and references to support “Pharmacological bioactivity results from the biodynamics and 35 biokinetics of a particular drug; these activities can be considered to form the drug space.”

Response 11: the following text was added: biodynamics characterizes the interaction between the drug and it is target, and biokinetics describes the path of the drug to it is target; thus, these two properties contain all systems involved in the drug-receptor interactions. The following references were added: 15. Austin Horton, Isaac T Schiefer. Pharmacokinetics and pharmacodynamics of nitric oxide mimetic agents. Nitric Oxide. 2019  1;84:69-78.

16.L D'Evoli, A Tarozzi, P Hrelia, M Lucarini, M Cocchiola, P Gabrielli, F Franco, F Morroni, G Cantelli-Forti, G Lombardi-Boccia. Influence of cultivation system on bioactive molecules synthesis in strawberries: spin-off on antioxidant and anti-proliferative activity. J Food Sci. 2010;75(1): C94-9.

Comment: 12) Introduction, p1: Justify and add references to support “Drug space is a particular region that is only a part of a vast region of the chemical space.”

The following references were added:

The following text was added: There is a vast number of distinct molecules. However, not all molecules have biological activity. So

  1. José L Medina-Franco , Karina Martinez-Mayorga, Nathalie Meurice. Balancing novelty with confined chemical space in modern drug discovery. Expert Opin Drug Discov. 2014;9(2):151-65.
  2. Giulio Vistoli , Alessandro Pedretti , Angelica Mazzolari , Bernard Testa. Approaching Pharmacological Space: Events and Components. Methods Mol Bio. 2018;1800:245-274.
  3. Konstantinos Pliakos , Celine Vens. Drug-target interaction prediction with tree-ensemble learning and output space reconstruction. BMC Bioinformatics. 2020; 7;21(1):49.

Comment: 13) Introduction, p1: Add explanations and references to “Chemical space is a conceptualization in cheminformatics that refers to the space including all possible molecules and chemical compounds defined by a given set of construction rules and boundary conditions.”

Answer 13: the following text was added: Therefore all possible compounds ( known and theoretical)  and chemical elements are components of chemical space

The following references were added:

  1. Mahendra Awale , Ricardo Visini, Daniel Probst, Josep Arús-Pous, Jean-Louis Reymond. Chemical Space: Big Data Challenge for Molecular Diversity. Chimia. 2017;71(10):661-666
  2. José L Medina-Franco, Ana L Chávez-Hernández, Edgar López-López, Fernanda I Saldívar-González. Chemical Multiverse: An Expanded View of Chemical Space. Mol Inform. 2022;41(11):e2200116.
  3. Torsten Hoffmann, Marcus Gastreich. The next level in chemical space navigation: going far beyond enumerable compound libraries. Drug Discov. Today. 2019;24(5):1148-1156.

Comment: 14) Introduction, p1: It was unclear what is the justification of 1060 molecules and how the number was selected. Add explanations and references to support “Theoretical chemical space in cheminformatics refers to the space of potentially pharmacologically active molecules with a size order of 1060 molecules.”

Response 14:  the following text was added :

1060 molecules also known as chemical reaction space

The following references were added :

  1. Sina Stocker, Gábor Csányi, Karsten Reuter, Johannes T Margraf. Machine learning in chemical reaction space. Nat Commun. 2020;11(1):5505.
  2. Asmaa Boufridi, Ronald J Quinn. Harnessing the Properties of Natural Products. Annu Rev Pharmacol Toxicol. 2018 ;58:451-470.

Comment: 15) Introduction, p1: Justify and add references to support “Exploration of chemical space can be performed computationally - in silica or experimentally – by chemical reactions.”

Answer 15: the following text was added Both processes in silica data mining or database serch and chemical reaction respectively return new compounds that finally lead to chemical space exploration

The following services were added: 25. Ying Yang , Kun Yao , Matthew P Repasky , Karl Leswing , Robert Abel , Brian K Shoichet , Steven V Jerome. Efficient Explo-ration of Chemical Space with Docking and Deep Learning. J Chem Theory Comput. 2021;17(11):7106-7119.

  1. Jan P Unsleber , Markus Reiher. The Exploration of Chemical Reaction Networks. Annu Rev Phys Chem. 2020;;71:121-142.

Comment: 16) Introduction, p2: Justify “Exploration of chemical space is performed by using in silica database of virtual molecules, visualized by projecting the multidimensional molecular property space in lower dimensions, characterized by physicochemical measurable/quantifiable properties [4].”

Anser 16: the foloowing thext was added ; As  stated before generation of valid chemical strucutures using any means ( experimental or in silica) lead to novel compounds 

The following references were added : 27. Adam Nelson , George Karageorgis. Natural product-informed exploration of chemical space to enable bioactive molecular discovery. RSC Med Chem. 2020 Dec 16;12(3):353-362.

  1. D-D Li , Y-P Hou, W Wang, H-L Zhu. Exploration of chemical space based on 4-anilinoquinazoline. Curr med Chem. 2012;19(6):871-92

Comment: 17) Introduction, p2: It was unclear why 12 electrons were selected. Justify and add references to support “Moving through the chemical space is performed by generating stoichiometric combinations of 12 electrons and atomic nuclei to approach all possible topology isomers to give onstruction rules.”

Response 17: the following references were added: all stoichiometric inorganic materials

The following text was added :

  1. Wenxi Zhao, Dmitriy Korobskiy, Shreya Chandrasekharan, Kenneth M Merz Jr, George Chacko. Converging Interests: Chemoinformatics, History, and Bibliometrics. J Chem Inf Model2020.60(12):5870-5872
  2. Daniel W Davies, Keith T Butler, Adam J Jackson, Andrew Morris, Jarvist M Frost , Jonathan M Skelton , Aron Walsh. Computational Screening of All Stoichiometric Inorganic Materials. Chem. 2016;1(4):617-627

Comment: 18) Introduction, p2: The sentence was too broad and unclear. Rephrase “In the real world, physicochemical properties are the projection of chemical properties in the known chemical space; being only projections, these properties are often not unique (i.e., degenerate), and different molecules can exhibit similar properties [6].”

Response18: the sentence was corrected: In the experimental situation real world, physicochemical properties are the projection of chemical properties in the known chemical space; being only projections, these properties are often not unique ( degenerate- i.e. different molecules with the same mass). Thus different molecules can exhibit similar properties [31].

Comment: 19) Introduction, p2: The sentence does not bring useful information. Delete “Materials design and drug discovery both involve the exploration of chemical space.”

Response 19: sentence deleted.

Comment: 20) Introduction, p2: It was unclear why benzoquinolines were selected and what are the problems to justify the investigation. Justify “In this work chemical space of benzoquinolines is explored. Topological space, chem- 58 ical space, and bioactivity space are discussed.”

Response 20: Being a bioactive and relatively easy-to-derivate molecule and considering previous published research – the chemical space of benzoquinolines was chosen for further research. 

Comment: 21) Results, p2: Specify which type of antimicrobial activity to support “Some of the obtained salts, namely3i, 3j, and 3n, have potent antimicrobial activity.”

Response 21: Corrected as suggested.

Comment: 22) Results, p2: Specify the specie origin of ATP synthase (microbial or mammalian ?)  and add references to support “From the obtained molecular docking studies, we noticed that salts 3i, 3j, and 3n have the best fit in complex with ATP synthase.”

Response 22: corrected

Comment: 23) Results, p2: Add explanations to support “We also noticed that ARG 357 shares electrons with heterocycle aromatic moiety [7].”

Response 23: corrected

Comment: 24) Results, p2: Specify BQS in “In the binding pocket of ATP synthase, a powerful hy- drogen bond between the oxygen atom (from CO group) and aminoacid ARG-364 is formed, stabilizing the complex salt BQS 3j - ATP synthase”

Response 24: corrected

Comment: 25) Results, Table 1: Specify abbreviations, A1,P2,R3,R4,R5,R6,R7 to support Table 1.

Response 25: corrected

Comment: 26) Results, add comments to support findings from Table 2, 3, and 4 in “Tables 2, 3, and 4 represent 78 QSAR models for bioactivities on S. aureus, E. coli, and C. Albicans.”

Response 26: there are no 78 QSAR models

Comment: 27) Results, p3: Explain what are the 5 descriptors,  why S. aureus was selected, it was unclear if the target enzyme was the ATP synthase to support “Multiple linear regression (MLR) QSAR model for S. aureus was build using the 5 descriptors set (see Table 2) and shows a Pearson correlation (r)= 0.972, Pearson correlation squared (r2)= 0.945704, Spearman rank correlation (p)= 0.964286, Mean squared devi- 84 ation (MSD)= 1.38611, Root mean square deviation (RMSD)= 1.17733 and Cross validated 85 square (q2)= 0.945704 respectively. The MLR equation is y= 0.9457x+0.8995 (Figure 1).”

Comment 27: corrected. The following text was added : ( targeting ATP synthase )was built using the five descriptors set (  steric energy- steric energy of the compound  kcal/mol; H bond -number of hydrogen bonds in the compound; heavy atoms-number of heavy atoms; N-number of nitrogen atoms, UnH bond- unbonded Hydrogen atoms;see Table 2)

Comment: 28) Results, p3: It was unclear how the percentage of the phenomena was computed and to which phenomena it referred. Justify, add explanations to support  “The  QSAR model shows that nitrogen atoms and unidirectional hydrogen bonds have a crucial role in the activity of S. aureus, explaining 69% of the phenomena. A heavy atom's presence significantly increases the bioactivity of the compound (r2= 0.90). Steric and hydrogen bond energy can explain less than 5 % of phenomena.”

Answer 28: corrected as suggested

Comment: 29) Results, p3: The sentence was unclear. It was unclear what are the ligands to support “Ligand composition is imperative inactivity on S. aureus, while ligand conformation and bonding do not significantly impact.”

Response 29: the sentence was corrected

Comment: 30) Results, p3: Indicate why E.Coli was selected and if the enzyme target was the ATP synthase to support “ultiple linear regression (MLR) QSAR model for E. coli build using 5 descriptors (see Table 3) shows a Pearson correlation (r)= 0.947, Pearson correlation squared (r2)= 0.896, Spearman rank correlation (p)= 0.896, Mean squared deviation (MSD)= 2.039, Root mean square deviation (RMSD)= 1.428 and Cross validated square (q2)= 0.896 respectively. The MLR equation is y= 0.8968x+1.3927 (Figure 1).”

Response 30: corrected

Comment: 31) Results, p3: Specify the ligand to support “The QSAR model suggests that the ligand structure has a critical role in the activity of E. coli.”

Response 31: ligand structures are shown in Scheme 1

Comment: 32) Results, p3: Add explanations to support “The degrees of freedom (DOF) 100 and N atoms can explain 70% of BQS3 bioactivity in E. coli. Also, the torsional energy of 101 the ligand halogen atom and unidirectional hydrogen bonds has been implied.”

Respnse 32: corrected

CommeRespont: 33) Results, p4: Indicate why C. Albicans was selected and if the enzyme target was the ATP synthasense to support “Multiple linear regression (MLR) QSAR model for C. Albicans build using 5 descriptors (Table 4) shows a Pearson correlation (r)= 0.9867, Pearson correlation squared  (r2)= 0.9737, Spearman rank correlation (p)= 0.964, Mean squared deviation (MSD)= 0.548, Root mean square deviation (RMSD)= 0.740 and Cross validated square (q2)= 0.975 respectively. The MLR equation is y= 0.973x+0.4678 (Figure 1).”

Response 33: correted

Comment: 34) Results, Fig 1, p4: Indicate in the Figure-1 legend what are the observed and predicted values to support “Figure 1. Scatter plot of observed (on ox axis) and predicted (on oy axis) MIC for: a. S. aureus; b. E. 115 coli, and c. C. albicans.”

Response 34: corrected

Comment: 35) Results, p4: It was unclear which activities and how predicted activities can be obtained to support “In Figure 1, scatter plots represent the observed and predicted activities on S. aureus, E. coli, and C. Albicans. Observed and predicted activities are roughly following each other.  One point is an outlier for S. aureus and E. coli due to lack of activity.”

Response 35: The QSAR methodology was described in the Material and methods section. The activities and predicted values have a correlation  coefficient  bigger than 0.9

Comment: 36) Results, p4: It was unclear from which data base were retrieved the 7000 compounds and if they were all benzoquinolines to support “The screening retrieved more than 7000 compounds that satisfy at least four criteria of the hypothesis A1P2R3R4R5R6R7; see supplemental files.”

Response 36: As stated in the methods section, the database was diverse. The pharmacophore used from the QSAT model of benzoquinolineqa was used for screening in the database. The result is listed in the supplemental files.

Comment: 37) Results, p5: The sentence was too broad. Specify types of activity and Justify “Screening results retrieved many heterocyclic compounds with nitrogen. These findings follow the computed QSAR model and suggest exploring a distinct class of compounds with superior activity on discussed strains.”

Response 37: the following text was added: As stated before, the screening template used was the pharmacophore hypothesis developed with the help of the QSAR model.

Comment: 38) Results, Figure 2, p6: Indicate the specie origin of the ATP synthase to support the Figure-2 legend in “Virtual screening cluster of compounds for BQS salts in interaction with ATP synthase is represented as a scatter plot: x- and y-axis represent coordinates of each compound characterized by its properties: relative energy, Phase vector score, phase volume score (see supplemental files).”

Response 39: corrected

Comment: 39) Results, p6: Specify the acceptor and the positive ionic groups to support “The pharmacophore data from Table 6 reveal the importance of the acceptor group 142 and the positive ionic group surrounded by aromatic centers.”

Response 39: corrected

Comment: 40) Results, p6: Specify activities in “Tables 7, 8, and 9 represent QSAR models for bioactivities on S. aureus, E. coli, and C. Albicans. ”

Response 40: corrected

Comment: 41) Results, p7: It was unclear to which type of ATP synthase the data from table 7 refers to in “Multiple linear regression (MLR) QSAR model for S. aureus was build using 5 descriptors (see Table 7) and shows a Pearson correlation (r)= 0.960, Pearson correlation squared (r2)= 0.921788, Spearman rank correlation (p)= 0.88, Mean squared deviation 149 (MSD)= 1.996, Root mean square deviation (RMSD)= 1.413 and Cross validated square 150 (q2)= 0.921788 respectively.”

Response 41: corrected

Comment: 42) Results, p7: It was unclear which type of bioactivities and which phenomena to support“QSAR models show that the presence of an N atom can explain 42% of the phenomena. The degrees of freedom also have a crucial role in bioactivity. Pose energy and torsion energy count for the rest of the phenomena. ”

Response 42: corrected

Comment: 43) Results, p7: It was unclear to which type of ATP synthase the data from table 8 refers to in “Multiple linear regression (MLR) QSAR model for E. coli build using the 5 descriptors a Pearson correlation (r)= 0.92798, Pearson correlation squared (r2)= 0.882256, Spearman rank correlation (p)= 0.91785, Mean squared deviation (MSD)= 2.32741, Root mean square deviation (RMSD)= 1.52559 and Cross validated square (q2)= 0.882256 respectively. The MLR equation is y= 0.8823x+1.5895 (Figure 4).”

Response 43: corrected

Comment: 44) Results, p7: Specify TOPO II in “The QSAR model suggests that the activity of E. coli of BQS 3 in a complex with TOPO II depends on the number of degrees of freedom (DOF), the number of nitrogen (N) atoms, and the torsional and total energy of the complex.”

Response 44: corrected

Comment: 45) Results, p7: It was unclear to which type of ATP synthase the data from table 9 refers to in “Multiple linear regression (MLR) QSAR model for C. albicans build using 5 descriptors (Table 9) shows a Pearson correlation (r)= 0.9867, Pearson correlation squared  (r2)= 0.9737, Spearman rank correlation (p)= 0.91785, Mean squared deviation (MSD)= 2.32741, Root mean square deviation (RMSD)= 1.52559 and Cross validated square (q2)=  0.882256 respectively. The MLR equation is y= 0.9737x+1.5895 (Figure 4).”

Response 45: corrected

Comment: 46) Results, p7: It was unclear how membrane permeabilities were assessed to support “QSAR model shows that bioactivity on C. Albicans of BQS salts three could be explained by the membrane permeability properties of the compounds. In addition, findings suggest that the polarizability of the ligand has a crucial role.”

Response 46: membrane operability was assessed by using membrane permeability descriptors. The following text was added: characterized by using membrane permeability descriptors -see Table 9).

Comment: 47) Results, Fig. 4, p8: Specify the observed and predicted values in the Figure-4 legend to support  “Figure 4. Scatter plot of observed (on ox axis) and predicted (on oy axis) MIC for a. S. aureus; b. E. 177 coli, and c. C. albicans.”

Response 47: corrected

Comment: 48) Results, p8: Specify which type of bioactivity to support “Overall the scatter plot demonstrates a correct prediction of the variable (bioactivity) by using the multiple linear regression 181 (MLR) QSAR models.”

Response 48: corrected            

Comment: 49) Results, p8: It was unclear from which data base was selected the 9000 compounds and why the previous findings were based from 7000 compounds to justify “The screening retrieved more than 9000 compounds that satisfy at least four criteria of the hypothesis A1P2R3R4R5R6R7; see supplemental files.”

Response 49: database was attached at the supplemental file

Comment: 50) Results, p8: Explain why topoisomerase II was selected and Indicate the specie origin of ATP synthase and Topoisomerase II. To support “Screening results show that compounds with heterocycles in structure and halogen 188 atoms bound with aromatic rings satisfy the hypothesis A1P2R3R4R5R6R7 and can be 189 active on ATP synthase and Topoisomerase II.”

Answer 50:  corrected. The following text was added: Both enzymes are crucial in bacterial grouth and multiplication, being preferable antibiotics targets.

Comment: 51) Discussion, p9: Indicate the specie origin of ATP synthase, and TOPO II. “Regarding screening compounds' structural properties, it is noted that both series of the compound screen against the two enemies, TOPO II and ATPase, have the same structural properties.”

Answerer 51: corrected

Comment: 52) Discussion, Fig 7, p10: Indicate the specie origin of ATP synthase, and TOPO II to support Figure-7 legend in . “Structural composition of molecular dataset resulted after screening”

Answer 52: corrected

Comment: 53) Materials and Methods, p11: Indicate the species of TOPO II and ATP-synthase“Using to support, indicate chemical structures of the two compounds to support  “Two essential compounds synthesized and tested [8], two single pharmacophore hypotheses were developed for best salts compounds with topoisomerase II (TOPO II ) and Adenosine triphosphate synthase (ATP-synthase) as targets.”

Response 53: corrected

Comment: 54) Materials and Methods, p11: It was unclear how the screening was performed and what was the target to support “An extensive virtual screening ( over 2 million compounds ) was performed using the resulting hypothesis against a diverse molecular data set. The data set used in the screening was retrieved from ChEMBL Database [10].”

Response 54: corrected

Comment: 55) Materials and Methods, p11: Indicate the specie of the enzymes in “TOPO II and ATP -synthase. PDB structures of TOPO II and ATP synthase were used.”

Response 55: corrected

Comment: 56) Conclusions, p12: It was unclear if compounds shall target the microbial enzymes or the human enzymes since there were insufficient information how biological activities were measerd. Rephrase, add information to support  “Screening results show that compounds with heterocycles in structure and halogen  atoms bound with aromatic rings satisfy the hypothesis A1P2R3R4R5R6R7 and can be active on ATP synthase and Topoisomerase II. Furthermore, antimicrobial and antitumoral activity seems to be related to the lipophilic properties of the compounds.”

Response 56: corrected.

Reviewer 2 Report

Lungu and co. represented the studies on QSAR model of benzoquinoline moiety and further explored for antibacterial and anticancer compounds via virtual screening as per model. The results are represented in well mannered form and provide some key findings yet the manuscript is very confusing and scattered for understanding of wide readership. Thus, the manuscript need to be potentially revised and answer some queries before final publication. 

1. The study started with benzoquinoline chemical space to for QSAR model but it revolved around further virtual screening of some other libraries without this moiety. What was purpose of selecting such libraries and how they are relevant to benzoquinoline moiety for further exploration. 

2. The introduction part is look like definitions of various terms rather than scope of study and research gap. It need to be potentially revised for scientific readership. 

3. Similarly, conclusion also does not define the findings of this studies and further scope of this work. It need to be relevant to study describing key findings and point from this work.

4. The conclusion should be present after discussion followed by material and method section.

5. The discussion part need to be explained in detail for findings. It is completed with only five sentences and two figures. Explain the results in detail with their validation and achievements of study in discussion part. 

6. Line 197-98: The sentence need to be removed from manuscript. The figure 6 is missing and after figure 5, it is directly to figure 7. 

7. Finally the manuscript require substantial major revision before further processing. 

Author Response

Lungu and co. represented the studies on QSAR model of benzoquinoline moiety and further explored for antibacterial and anticancer compounds via virtual screening as per model. The results are represented in well mannered form and provide some key findings yet the manuscript is very confusing and scattered for understanding of wide readership. Thus, the manuscript need to be potentially revised and answer some queries before final publication.

Comment: 1. The study started with benzoquinoline chemical space to for QSAR model but it revolved around further virtual screening of some other libraries without this moiety. What was purpose of selecting such libraries and how they are relevant to benzoquinoline moiety for further exploration.

Response 1: The pharmacophore hypothesis resulting from the QSAr models of benzoquinones was used for further screening to expand and further explore the benzoquinone chemical space. Using a pharmacophore hypothesis with seven structural futures ( A1P2R3R4R5R6R7) preamble compound with similar activities will result. Although these compounds will suggest further functionalization of benzoquinone moiety.

Comment 2 The introduction part is look like definitions of various terms rather than scope of study and research gap. It needs to be potentially revised for scientific readership.

Response 2: The introduction was rephrased.

Comment:3 Similarly, conclusion also does not define the findings of this studies and further scope of this work. It need to be relevant to study describing key findings and point from this work.

Response3:corrected

Comment4 The conclusion should be present after discussion followed by material and method section.

Response 4: the manuscript was organized after  the journal template.

Comment 5 The discussion part need to be explained in detail for findings. It is completed with only five sentences and two figures. Explain the results in detail with their validation and achievements of study in discussion part.

Response 5:the  entire discussion part was rephrased and reorganized:

 Chemical or molecular similarity refers to the similarity of chemical elements, molecules, or chemi-cal compounds concerning functional qualities. Biological effects are quantified using the biologi-cal activity of a compound. The function can be related to the chemical activity of a  compound. The chemical similarity is often described as an inverse of a distance measure in descriptor space (i.e.for inverse distance measures are the molecule kernels)[33]. The concept is essential in cheminformatics.[34] It plays a vital role in approaches to predicting the properties of chemical compounds, designing molecules with predefined properties, and, especially, conducting drug de-sign studies by screening large databases containing structures of available (or potentially availa-ble) chemicals. These concepts are based on the similar property principle of Johnson and Maggio-ra, which states: that similar compounds have similar properties[35]. The similarity-based[36] vir-tual screening states that all compounds in a database similar to a query compound have similar biological activity. However, this hypothesis is not always valid, while the set of retrieved com-pounds is often considerably enriched with actives[37,38]. Molecular structures are usually repre-sented by structural keys or fixed or variable-size molecular fingerprints. Molecular screens and fingerprints can contain various two-dimensional or three-dimensional information.

Chemical similarity can be expanded to similarity network theory in intense, extensive library vir-tual screening clusters of compounds ( Figure 8). The theory state that descriptive network prop-erties and graph theory can be applied to analyze sample chemical space, estimate chemical di-versity and predict drug target. Also, 3D chemical similarity networks based on 3D ligand con-formation have been developed, which can be used to identify scaffold-hopping ligands. Cluster analysis resulting after docking screening against bacterial TOPO II and  ATPase retrieves one cluster for both enzymes with relatively no outliers (Figure 8). These findings suggest a common chemical space for both bioac-tivities.

Comment:6 Line 197-98: The sentence need to be removed from manuscript. The figure 6 is missing, and after figure 5, it is directly to figure 7.

Response 6: the sentence was removed

Comment:7 Finally, the manuscript require substantial major revision before further processing.

Response 7: the manuscript was substantially corrected.

Reviewer 3 Report

In this manuscript authors demonstrated the chemical space of benzoquinolines as well as topological space, chemical space, and bioactivity space are explored. The manuscript suitable for molecules journal and some minor concerns should be consider before its publication. The language needs to be improved aa well as conditions of the schemes and in detail explanation of tables and figures should be included. the references style must be uniform in all cases. 

Author Response

In this manuscript authors demonstrated the chemical space of benzoquinolines as well as topological space, chemical space, and bioactivity space are explored. The manuscript suitable for molecules journal and some minor concerns should be consider before its publication.

Comment1: The language needs to be improved aa well as conditions of the schemes and in detail explanation of tables and figures should be included.

Response1: the language was improved and the figure legend was further expanded

Comment2:  the references style must be uniform in all cases. 

Response 2: the reference style was corrected. 

Round 2

Reviewer 1 Report

The abstract was too general and did not indicate clearly the aims and the main findings The introduction did not justify the research performed. It was unclear why benzoquinolines were selected and which problems remain to be solved. Several sentences lacked justification and references. There was insufficient information what was the target of benzoquinolines. It was unclear if benzoquinolines shall target microbial or mammalian enzymes. There were no explanations how microbial activities (and which ones) could be assessed. The overall manuscript was too broad and did not add significant information, as reflected in the discussion and in the conclusion. 

Reviewer’s answer: 

The abstract remained too broad. The drug target was unspecified, the problems were not delineated and the aims remain undefined.

The overall manuscript did not add significant information, as reflected in the discussion and in the conclusion. The two eventual putative compounds were not tested for their biological activities. 

Comment: 2) Abstract, p1: Avoid general statements. Delineate exactly the research aims. Specify drug targets and Rephrase “This computational study emphasizes benzoquinone drug space.”

Response 2: corrected as suggested> The following phrases were added: This computational study aims to characterize e benzoquinone drug space to ease future drug design processes based on these molecules. 

Reviewer’s answer: It was insufficiently addressed. Specify drug space and identify drug target.

Comment:3) Abstract, p1: The sentence was unclear. Replace “virtual” with another word. Specify topological, chemical and bioactivity spaces to support “Topological, chemical, and bioactivity spaces are explored using virtual methodologies based on virtual screening and scaffold hopping.”

Response 3: corrected as suggested. Virtual was replaced with computational.

Reviewer’s answer: It was insufficiently addressed. It remained too broad. Specify chemical (which groups), bioactivity (which ones), methodologies (which ones). 

Comment:4) Abstract, p1: add more information to support “Results show a poly-morphological chemical space 16 that suggests distinct characteristics.”

Reviewer’s answer: It was insufficiently addressed. It remained too broad. The meaning of poly-morphological chemical space is still unclear. The characteristics were not specified. 

Response 4: The following phrases were added: The chemical space seems to be correlated with properties like steric energy, the number of hydrogen bonds, the presence of halogen atoms, and membrane permeability-related properties. 

Reviewer’s answer: Avoid speculative statement. Is it correlated or not? 

Comment: 5) Abstract, p1: Replace “novel” by chemical names, and identify what are the drug targets. “Lastly, novel chemical compounds with drug-like potential have been identified.”

Reviewer’s answer: Several journals discourage the use of “novel”. 

Response 5 : correted . The following line was added: Lastly, novel chemical compounds with drug-like potential have been identified, like oxadiazole methybenzamide and floro methylcyclohexane diene. 

Reviewer’s answer: Several journals discourage the use of “novel”. Specify drug targets.

Comment: 9) Introduction, p1: Add explanations and references to support “Manifold spaces may be weighted to,, carry" some properties, such as bioactivity.”

Response 9: the following line was added, and their references: Particularly receptor ligandin interaction carries a vast range of properties. Those interactions can be described as stated before by manifllds [12,13]. 

  1. Ken J Ishii, Cevayir Coban, Shizuo Akira. Manifold mechanisms of Toll-like receptor-ligand recognition. J Clinical Imunolo-gy. 2005;25(6):511-21.
  2. Andras Piffko , Christian Uhl , Peter Vajkoczy , Marcus Czabanka , Thomas Broggini. EphrinB2-EphB4 Signaling in Neuroon-cological Disease. Int J Mol Sci. 2022 Jan 31;23(3):1679.

Reviewer’s answer: The updated sentence was unclear. Specify which properties, and  interactions. What is  the meaning of  “manifllds”

Comment: 10) Introduction, p1: Add explanations and references to support “Thus a variable describing the bioactivity can be represented using manifolds.”

Response 10: the following reference was added

  1. Vince M Lombardo , Christopher D Thomas, Karl A Scheidt. A tandem isomerization/prints strategy: iridium(III)/Brønsted acid cooperative catalysis. Angew Chem Int Ed 2013;52(49):12910-4.

Reviewer’s answer: Add explanations.

Comment: 20) Introduction, p2: It was unclear why benzoquinolines were selected and what are the problems to justify the investigation. Justify “In this work chemical space of benzoquinolines is explored. Topological space, chemical space, and bioactivity space are discussed.”

Response 20: Being a bioactive and relatively easy-to-derivate molecule and considering previous published research – the chemical space of benzoquinolines was chosen for further research. 

Reviewer’s answer: What are the problems to justify the research? 

Comment: 21) Results, p2: Specify which type of antimicrobial activity to support “Some of the obtained salts, namely3i, 3j, and 3n, have potent antimicrobial activity.”

Response 21: Corrected as suggested.

Reviewer’s answer: The antimicrobial activity was not specified. If the drug target is the ATP synthase, it is the ATP synthase activity that shall be determined. The antimicrobial activity could be something unrelated to the ATP synthase. That is why it is necessary to provide exact information on how antimicrobial activity is performed. Omitting this information may impact the findings.  

Comment: 22) Results, p2: Specify the specie origin of ATP synthase (microbial or mammalian ?) and add references to support “From the obtained molecular docking studies, we noticed that salts 3i, 3j, and 3n have the best fit in complex with ATP synthase.”

Response 22: corrected

Reviewer’s answer: That is correct. This shall be microbial ATP synthase. However, it was unclear if the docking was performed on microbial ATP synthase and which one ? 

Comment: 23) Results, p2: Add explanations to support “We also noticed that ARG 357 shares electrons with heterocycle aromatic moiety [7].”

Response 23: corrected

Reviewer’s answer: It was unclear if the docking was performed on microbial ATP synthase and which one ? Check reference 32 (in the manuscript) since it seems unrelated to the sentence. 

Comment: 24) Results, p2: Specify BQS in “In the binding pocket of ATP synthase, a powerful hy- drogen bond between the oxygen atom (from CO group) and aminoacid ARG-364 is formed, stabilizing the complex salt BQS 3j - ATP synthase”

Response 24: corrected

Reviewer’s answer: It was unclear if the docking was performed on microbial ATP synthase and which one ? 

Comment: 25) Results, Table 1: Specify abbreviations, A1,P2,R3,R4,R5,R6,R7 to support Table 1.

Response 25: corrected

Reviewer’s answer: It remained unspecified.  

Comment: 26) Results, add comments to support findings from Table 2, 3, and 4 in “Tables 2, 3, and 4 represent 78 QSAR models for bioactivities on S. aureus, E. coli, and C. Albicans.”

Response 26: there are no 78 QSAR models

Reviewer’s answer: Sorry for the mistake. It is not “78 QSAR models” but “QSAR models”. Nevertheless, some comments are needed. 

Comment: 27) Results, p3: Explain what are the 5 descriptors, why S. aureus was selected, it was unclear if the target enzyme was the ATP synthase to support “Multiple linear regression (MLR) QSAR model for S. aureus was build using the 5 descriptors set (see Table 2) and shows a Pearson correlation (r)= 0.972, Pearson correlation squared (r2)= 0.945704, Spearman rank correlation (p)= 0.964286, Mean squared devi- 84 ation (MSD)= 1.38611, Root mean square deviation (RMSD)= 1.17733 and Cross validated 85 square (q2)= 0.945704 respectively. The MLR equation is y= 0.9457x+0.8995 (Figure 1).”

Comment 27: corrected. The following text was added : ( targeting ATP synthase )was built using the five descriptors set ( steric energy- steric energy of the compound kcal/mol; H bond -number of hydrogen bonds in the compound; heavy atoms-number of heavy atoms; N-number of nitrogen atoms, UnH bond- unbonded Hydrogen atoms;see Table 2)

Reviewer’s answer: Indicate if S. aureus ATP synthase was used to build the five descriptors (PDB accessing number and reference for S. aureus ATP synthase are needed). 

Comment: 28) Results, p3: It was unclear how the percentage of the phenomena was computed and to which phenomena it referred. Justify, add explanations to support “The QSAR model shows that nitrogen atoms and unidirectional hydrogen bonds have a crucial role in the activity of S. aureus, explaining 69% of the phenomena. A heavy atom's presence significantly increases the bioactivity of the compound (r2= 0.90). Steric and hydrogen bond energy can explain less than 5 % of phenomena.”

Answer 28: corrected as suggested

Reviewer’s answer: It is still unclear if the phenomena is directly associated to the ATP synthase activity. 

Comment: 29) Results, p3: The sentence was unclear. It was unclear what are the ligands to support “Ligand composition is imperative inactivity on S. aureus, while ligand conformation and bonding do not significantly impact.”

Response 29: the sentence was corrected

Reviewer’s answer: It is still unclear what are the ligands. 

Comment: 30) Results, p3: Indicate why E.Coli was selected and if the enzyme target was the ATP synthase to support “ultiple linear regression (MLR) QSAR model for E. coli build using 5 descriptors (see Table 3) shows a Pearson correlation (r)= 0.947, Pearson correlation squared (r2)= 0.896, Spearman rank correlation (p)= 0.896, Mean squared deviation (MSD)= 2.039, Root mean square deviation (RMSD)= 1.428 and Cross validated square (q2)= 0.896 respectively. The MLR equation is y= 0.8968x+1.3927 (Figure 1).”

Response 30: corrected

Reviewer’s answer: It is still unclear Indicate if E. coli ATP synthase was used to build the five descriptors (PDB  accessing number and reference for E. coli ATP synthase are needed).

Comment: 31) Results, p3: Specify the ligand to support “The QSAR model suggests that the ligand structure has a critical role in the activity of E. coli.”

Response 31: ligand structures are shown in Scheme 1

Reviewer’s answer. Add this information in the manuscript. 

Comment: 32) Results, p3: Add explanations to support “The degrees of freedom (DOF) 100 and N atoms can explain 70% of BQS3 bioactivity in E. coli. Also, the torsional energy of 101 the ligand halogen atom and unidirectional hydrogen bonds has been implied.”

Respnse 32: corrected

Reviewer’s answer. Add more explanations.  

CommeRespont: 33) Results, p4: Indicate why C. Albicans was selected and if the enzyme target was the ATP synthasense to support “Multiple linear regression (MLR) QSAR model for C. Albicans build using 5 descriptors (Table 4) shows a Pearson correlation (r)= 0.9867, Pearson correlation squared (r2)= 0.9737, Spearman rank correlation (p)= 0.964, Mean squared deviation (MSD)= 0.548, Root mean square deviation (RMSD)= 0.740 and Cross validated square (q2)= 0.975 respectively. The MLR equation is y= 0.973x+0.4678 (Figure 1).”

Response 33: correted

Reviewer’s answer: It is still unclear Indicate if C. Albicans ATP synthase was used to build the five descriptors (PDB  accessing number and reference for C. Albicans ATP synthase are needed).

Comment: 34) Results, Fig 1, p4: Indicate in the Figure-1 legend what are the observed and predicted values to support “Figure 1. Scatter plot of observed (on ox axis) and predicted (on oy axis) MIC for: a. S. aureus; b. E. 115 coli, and c. C. albicans.”

Response 34: corrected

Reviewer’s answer: It is insufficient. It is unclear if the minimum inhibitory concentrations referred to the ATP synthase activities or to antimicrobial activities.

Comment: 35) Results, p4: It was unclear which activities and how predicted activities can be obtained to support “In Figure 1, scatter plots represent the observed and predicted activities on S. aureus, E. coli, and C. Albicans. Observed and predicted activities are roughly following each other. One point is an outlier for S. aureus and E. coli due to lack of activity.”

Response 35: The QSAR methodology was described in the Material and methods section. The activities and predicted values have a correlation coefficient bigger than 0.9

Reviewer’s answer: It is insufficient. It is unclear if the minimum inhibitory concentrations referred to the ATP synthase activities or to antimicrobial activities. 

Comment: 37) Results, p5: The sentence was too broad. Specify types of activity and Justify “Screening results retrieved many heterocyclic compounds with nitrogen. These findings follow the computed QSAR model and suggest exploring a distinct class of compounds with superior activity on discussed strains.”

Response 37: the following text was added: As stated before, the screening template used was the pharmacophore hypothesis developed with the help of the QSAR model.

Reviewer’s answer: The type of activity remained unclear. For the sake of clarity: Is that ATP synthase activity or is that antimicrobial activity ? No comparisons can be performed with antimicrobial activity since it may not be associated to a ATP synthase activity. The drug may act on a distinct target than ATP synthase if a microbial activity is measured. 

Comment: 38) Results, Figure 2, p6: Indicate the specie origin of the ATP synthase to support the Figure-2 legend in “Virtual screening cluster of compounds for BQS salts in interaction with ATP synthase is represented as a scatter plot: x- and y-axis represent coordinates of each compound characterized by its properties: relative energy, Phase vector score, phase volume score (see supplemental files).”

Response 39: corrected

Reviewer’s answer: Not answered. Avoid broad sentences, indicate clearly specie origin for the calculations used (PDB of each microbial specy). 

Comment: 40) Results, p6: Specify activities in “Tables 7, 8, and 9 represent QSAR models for bioactivities on S. aureus, E. coli, and C. Albicans. ”

Response 40: corrected

Reviewer’s answer: Not answered. indicate clearly types of activities. Is that ATP synthase activity ? 

Comment: 41) Results, p7: It was unclear to which type of ATP synthase the data from table 7 refers to in “Multiple linear regression (MLR) QSAR model for S. aureus was build using 5 descriptors (see Table 7) and shows a Pearson correlation (r)= 0.960, Pearson correlation squared (r2)= 0.921788, Spearman rank correlation (p)= 0.88, Mean squared deviation 149 (MSD)= 1.996, Root mean square deviation (RMSD)= 1.413 and Cross validated square 150 (q2)= 0.921788 respectively.”

Response 41: corrected

Reviewer’s answer: No comparisons can be performed with antimicrobial activity since it may not be associated to the ATP synthase activity. The drug may act on a distinct target than ATP synthase if a microbial activity is measured.

Comment: 42) Results, p7: It was unclear which type of bioactivities and which phenomena to support“QSAR models show that the presence of an N atom can explain 42% of the phenomena. The degrees of freedom also have a crucial role in bioactivity. Pose energy and torsion energy count for the rest of the phenomena. ”

Response 42: corrected

Reviewer’s answer: No comparisons can be performed with antimicrobial activity since it may not be associated to the ATP synthase activity. The drug may act on a distinct target than ATP synthase if a microbial activity is measured.

Comment: 43) Results, p7: It was unclear to which type of ATP synthase the data from table 8 refers to in “Multiple linear regression (MLR) QSAR model for E. coli build using the 5 descriptors a Pearson correlation (r)= 0.92798, Pearson correlation squared (r2)= 0.882256, Spearman rank correlation (p)= 0.91785, Mean squared deviation (MSD)= 2.32741, Root mean square deviation (RMSD)= 1.52559 and Cross validated square (q2)= 0.882256 respectively. The MLR equation is y= 0.8823x+1.5895 (Figure 4).”

Response 43: corrected

Reviewer’s answer: No comparisons can be performed with microbial activity since it may not be associated to the ATP synthase activity. The drug may act on a distinct target than ATP synthase if a antimicrobial activity is measured.

Comment: 45) Results, p7: It was unclear to which type of ATP synthase the data from table 9 refers to in “Multiple linear regression (MLR) QSAR model for C. albicans build using 5 descriptors (Table 9) shows a Pearson correlation (r)= 0.9867, Pearson correlation squared (r2)= 0.9737, Spearman rank correlation (p)= 0.91785, Mean squared deviation (MSD)= 2.32741, Root mean square deviation (RMSD)= 1.52559 and Cross validated square (q2)= 0.882256 respectively. The MLR equation is y= 0.9737x+1.5895 (Figure 4).”

Response 45: corrected

Reviewer’s answer: No comparisons can be performed with microbial activity since it may not be associated to the ATP synthase activity. The drug may act on a distinct target than ATP synthase if a antimicrobial activity is measured.

Comment: 46) Results, p7: It was unclear how membrane permeabilities were assessed to support “QSAR model shows that bioactivity on C. Albicans of BQS salts three could be explained by the membrane permeability properties of the compounds. In addition, findings suggest that the polarizability of the ligand has a crucial role.”

Response 46: membrane operability was assessed by using membrane permeability descriptors. The following text was added: characterized by using membrane permeability descriptors -see Table 9).

Reviewer’s answer: It is still unclear how membrane permeabilities were determined. Table 9 apparently did not report any membrane permeability data.

Comment: 47) Results, Fig. 4, p8: Specify the observed and predicted values in the Figure-4 legend to support “Figure 4. Scatter plot of observed (on ox axis) and predicted (on oy axis) MIC for a. S. aureus; b. E. 177 coli, and c. C. albicans.”

Response 47: corrected

Comment: 48) Results, p8: Specify which type of bioactivity to support “Overall the scatter plot demonstrates a correct prediction of the variable (bioactivity) by using the multiple linear regression 181 (MLR) QSAR models.”

Response 48: corrected 

Reviewer’s answer: Not answered. indicate clearly types of activities. Is that ATP synthase activity ? 

Comment: 50) Results, p8: Explain why topoisomerase II was selected and Indicate the specie origin of ATP synthase and Topoisomerase II. To support “Screening results show that compounds with heterocycles in structure and halogen 188 atoms bound with aromatic rings satisfy the hypothesis A1P2R3R4R5R6R7 and can be 189 active on ATP synthase and Topoisomerase II.”

Answer 50: corrected. The following text was added: Both enzymes are crucial in bacterial grouth and multiplication, being preferable antibiotics targets.

Reviewer’s answer: indicate if ATP synthase activity of Topoisomerase II was targeted ?

Comment: 51) Discussion, p9: Indicate the specie origin of ATP synthase, and TOPO II. “Regarding screening compounds' structural properties, it is noted that both series of the compound screen against the two enemies, TOPO II and ATPase, have the same structural properties.”

Answerer 51: corrected

Reviewer’s answer: indicate if ATP synthase activity of Topoisomerase II was targeted ?

Comment: 52) Discussion, Fig 7, p10: Indicate the specie origin of ATP synthase, and TOPO II to support Figure-7 legend in . “Structural composition of molecular dataset resulted after screening”

Answer 52: corrected

Reviewer’s answer: It is unclear  if ATP synthase activity was determined. 

Comment: 53) Materials and Methods, p11: Indicate the species of TOPO II and ATP-synthase“Using to support, indicate chemical structures of the two compounds to support “Two essential compounds synthesized and tested [8], two single pharmacophore hypotheses were developed for best salts compounds with topoisomerase II (TOPO II ) and Adenosine triphosphate synthase (ATP-synthase) as targets.”

Response 53: corrected

Reviewer’s answer: It remains unspecified. I have to dig from the previous papers of the authors to find an answer.  From reference: Vasilichia Antoci et al.  Benzoquinoline Derivatives: A Straightforward and Efficient Route to Antibacterial and Antifungal Agents. Pharmaceuticals 2021, 14(4), 335; https://doi.org/10.3390/ph14040335, it appears that human V-ATPase and not bacterial ATPase was used for docking. The docking and theoretical predictions based from the human V-ATP synthase cann’t be used for microbial ATP synthase. Poisons targeting human ATP synthase  instead of drugs targeting microbial ATP synthase are being screened. 

Comment: 55) Materials and Methods, p11: Indicate the specie of the enzymes in “TOPO II and ATP -synthase. PDB structures of TOPO II and ATP synthase were used.”

Reviewer’s answer: It remains unspecified. See comment 53. 

Response 55: corrected

Comment: 56) Conclusions, p12: It was unclear if compounds shall target the microbial enzymes or the human enzymes since there were insufficient information how biological activities were measerd. Rephrase, add information to support “Screening results show that compounds with heterocycles in structure and halogen atoms bound with aromatic rings satisfy the hypothesis A1P2R3R4R5R6R7 and can be active on ATP synthase and Topoisomerase II. Furthermore, antimicrobial and antitumoral activity seems to be related to the lipophilic properties of the compounds.”

Response 56: corrected.

Reviewer’s answer: It remains unspecified. See comment 53. 

Author Response

The abstract was too general and did not clearly indicate the aims and the main findings The introduction did not justify the research performed. It was unclear why benzoquinolines were selected and which problems remain to be solved. Several sentences lacked justification and references. There was insufficient information what was the target of benzoquinolines. It was unclear if benzoquinolines shall target microbial or mammalian enzymes. There were no explanations for how microbial activities (and which ones) could be assessed. The overall manuscript was too broad and did not add significant information, as reflected in the discussion and in conclusion.

Reviewer's answer:

Comment1: The abstract remained too broad. The drug target was unspecified, the problems were not delineated and the aims remained undefined.

The overall manuscript did not add significant information, as reflected in the discussion and in the conclusion. The two eventual putative compounds were not tested for their biological activities.

 Response 1: The manuscript was significantly improved to clarify the study's purpose. Drug targets have been clearly specified. There are no compounds to be tested. This is a computational study that explores the benzoquinoline chemical space.

Comment: 2) Abstract, p1: Avoid general statements. Delineate exactly the research aims. Specify drug targets and Rephrase "This computational study emphasizes benzoquinone drug space."

Response 2: corrected as suggested> The following phrases were added: This computational study aims to characterize e benzoquinone drug space to ease future drug design processes based on these molecules.

Reviewer's answer: It was insufficiently addressed. Specify drug space and identify drug target.

Response 2: the following phrase was added: The drug space is composed of all benzoquinone  that is active on topoisomerase II and ATP synthase

Comment:3) Abstract, p1: The sentence was unclear. Replace "virtual" with another word. Specify topological, chemical and bioactivity spaces to support "Topological, chemical, and bioactivity spaces are explored using virtual methodologies based on virtual screening and scaffold hopping."

Response 3: corrected as suggested. Virtual was replaced with computational.

Reviewer's answer: It was insufficiently addressed. It remained too broad. Specify chemical (which groups), bioactivity (which ones), and methodologies (which ones).

 Response 3: the following text was added: Topological space is a geometrical space in which the elements composing it can be defined as a set of neighbors (which satisfy a particular axiom). In such space, a chemical space can be defined as the property space spanned by all possible molecules and chemical compounds adhering to a given set of construction principles and boundary conditions. The potentially pharmacologically active molecules form the bioactivity space in this chemical space.

Comment:4) Abstract, p1: add more information to support "Results show a poly-morphological chemical space 16 that suggests distinct characteristics."

Reviewer's answer: It was insufficiently addressed. It remained too broad. The meaning of poly-morphological chemical space is still unclear. The characteristics were not specified.

Response 4: The following phrases were added: The chemical space seems to be correlated with properties like steric energy, the number of hydrogen bonds, the presence of halogen atoms, and membrane permeability-related properties.

Reviewer's answer: Avoid speculative statement. Is it correlated or not?

Response 4: the following correction was performed: the chemical space is correlated.

Comment: 5) Abstract, p1: Replace "novel" by chemical names, and identify what are the drug targets. "Lastly, novel chemical compounds with drug-like potential have been identified."

Reviewer's answer: Several journals discourage the use of "novel".

Response 5 : correted . The following line was added: Lastly, novel chemical compounds with drug-like potential have been identified, like oxadiazole methybenzamide and floro methylcyclohexane diene.

Reviewer's answer: Several journals discourage the use of "novel". Specify drug targets.

Response 5: the last phrase was rewritten: Lastly, novel chemical compounds(like oxadiazole methybenzamide and floro methylcyclohexane diene)  with drug-like poten-tial, active on TOPO II and ATP synthase have been identified.

Comment: 9) Introduction, p1: Add explanations and references to support "Manifold spaces may be weighted to,, carry" some properties, such as bioactivity."

Response 9: the following line was added, and their references: Particularly receptor ligandin interaction carries a vast range of properties. Those interactions can be described as stated before by manifllds [12,13].

Ken J Ishii, Cevayir Coban, Shizuo Akira. Manifold mechanisms of Toll-like receptor-ligand recognition. J Clinical Imunolo-gy. 2005;25(6):511-21.

Andras Piffko , Christian Uhl , Peter Vajkoczy , Marcus Czabanka , Thomas Broggini. EphrinB2-EphB4 Signaling in Neuroon-cological Disease. Int J Mol Sci. 2022 Jan 31;23(3):1679.

 Reviewer's answer: The updated sentence was unclear. Specify which properties, and  interactions. What is  the meaning of "manifolds"

Response 9: the following text was added: A manifold is pivotal to many parts of geometry because it allows complicated structures to be described in terms of well-understood topological properties of elementary spaces. Manifolds naturally arise as solution sets of systems of equations and as graphs of functions,. Such properties can be ligand-receptor interaction energies, ligand conformational energies, and lipophilic-related properties, respectively,,

Comment: 10) Introduction, p1: Add explanations and references to support "Thus a variable describing the bioactivity can be represented using manifolds."

Response 10: the following reference was added

Vince M Lombardo , Christopher D Thomas, Karl A Scheidt. A tandem isomerization/prints strategy: iridium(III)/Brønsted acid cooperative catalysis. Angew Chem Int Ed 2013;52(49):12910-4.

Reviewer's answer: Add explanations.

Response10: the following text was added: (i.e, by computing a set of equations that describe or are related to  the variable)

 Comment: 20) Introduction, p2: It was unclear why benzoquinolines were selected and what are the problems to justify the investigation. Justify "In this work chemical space of benzoquinolines is explored. Topological space, chemical space, and bioactivity space are discussed."

Response 20: Being a bioactive and relatively easy-to-derivate molecule and considering previous published research, benzoquinolines' chemical space was chosen for further research.

Reviewer's answer: What are the problems to justify the research?

The following text was added: Infectious diseases significantly cause morbidity and mortality, which can worsen the current antimicrobial resistance crisis.[32] Antimicrobial resistance is evolving rapidly. In addition, antimicrobials are valuable resources that enhance the prevention and treat-ment of infections. As resistance diminishes this resource, it is a societal goal to minimize resistance and reduce forces that produce resistance. Genetic recombination allows bacteria to rapidly disseminate genes encoding for antimicrobial resistance within and across species.[33] Antimicrobial use creates a selective evolutionary pressure, which leads to further resistance. Antimicrobial stewardship, best use, and infection prevention are the most effective ways to slow the spread and development of antimicrobial resistance [34,35]. Thus developing new drugs is crucial in managing antibiotic resistance.[36,37]

 Response 20: the following references were added : 32. Scott A McEwen , Peter J Collignon. Antimicrobial Resistance: a One Health Perspective. Microbial Spectr. 2018;6(2).

  1. Maurizio Ferri, Elena Ranucci, Paola Romagnoli, Valerio Giaccone. Antimicrobial resistance: A global emerging threat to public health systems. Crit Rev Food Sci Nutr. 2017 Sep 2;57(13):2857-2876.
  2. Carl Nathan. Resisting antimicrobial resistance. Nat Rev Microbial. 2020 ;18(5):259-260.
  3. Gang Liu , Line Elnif Thomsen, John Elmerdahl Olsen. Antimicrobial-induced horizontal transfer of antimicrobial resistance genes in bac-teria: a mini-review. J Antimicrob Chemother. 2022;77(3):556-567.
  4. Ritika Kabra, Nutan Chauhan, Anurag Kumar, Prajakta Ingale, Shailza Singh. Efflux pumps and antimicrobial resistance: Paradoxical components in systems genomics. Prog Biophys Mol Biol. 2019 Jan;141:15-24.
  5. Ines Mack, Julia Bielicki. What Can We Do About Antimicrobial Resistance? Pediatr Infect Dis J. 2019 Jun;38(6S Suppl 1):S33-S38.

Comment: 21) Results, p2: Specify which type of antimicrobial activity to support "Some of the obtained salts, namely3i, 3j, and 3n, have potent antimicrobial activity."

Response 21: Corrected as suggested.

Reviewer's answer: The antimicrobial activity was not specified. If the drug target is the ATP synthase, it is the ATP synthase activity that shall be determined. The antimicrobial activity could be something unrelated to the ATP synthase. That is why it is necessary to provide exact information on how antimicrobial activity is performed. Omitting this information may impact the findings. 

Response 21: the following text was added: Some of the obtained salts, namely 3i, 3j, and 3n, have potent antimicrobial activity on S. aureus, E. coli, and C. Albicans strains.

 Comment: 22) Results, p2: Specify the specie origin of ATP synthase (microbial or mammalian ?) and add references to support "From the obtained molecular docking studies, we noticed that salts 3i, 3j, and 3n have the best fit in complex with ATP synthase."

Response 22: corrected

Reviewer's answer: That is correct. This shall be microbial ATP synthase. However, it was unclear if the docking was performed on microbial ATP synthase and which one?

Response 22: the following text was added at the material and methods section: The cristal structures used for docking studies were 1AB4 for TOPO II[39]  and 1EF0 for ATP synthase[40]

The references were added: 39. Cabral, J.H., Jackson, A.P., Smith, C.V., Shikotra, N., Maxwell, A., Liddington, R.C. Crystal structure of the breakage-reunion domain of DNA gyrase. Nature (1997) 388: 903-906.

  1. Poland, B.W., Xu, M.Q., Quiocho, F.A. Structural insights into the protein splicing mechanism of PI-SceIJ Biol Chem. 2000;275: 16408-16413

Comment: 23) Results, p2: Add explanations to support "We also noticed that ARG 357 shares electrons with heterocycle aromatic moiety [7]."

Response 23: corrected

Reviewer's answer: It was unclear if the docking was performed on microbial ATP synthase and which one ? Check reference 32 (in the manuscript) since it seems unrelated to the sentence.

Response 23: reference 32 was replaced Yinfeng Yang , Yan Li , Yanqiu Pan , Jinghui Wang , Feng Lin , Chao Wang , Shuwei Zhang , Ling Yang. Computational Analysis of Structure-Based Interactions for Novel H₁-Antihistamines. Int J Mol Sci 2016 Jan 19;17(1):129.

The following text was added: . From the docking studies by using ligand interaction viewing dialog, we also noticed that ARG 357 shares electrons with heterocycle aromatic moiety [32].

 Comment: 24) Results, p2: Specify BQS in "In the binding pocket of ATP synthase, a powerful hy- drogen bond between the oxygen atom (from CO group) and aminoacid ARG-364 is formed, stabilizing the complex salt BQS 3j - ATP synthase"

Response 24: corrected

Reviewer's answer: It was unclear if the docking was performed on microbial ATP synthase and which one ?

Response 24: corrected ( microbial) ( see material and methods)

Comment: 25) Results, Table 1: Specify abbreviations, A1,P2,R3,R4,R5,R6,R7 to support Table 1.

Respose 25: corrected

Reviewer's answer: It remained unspecified. 

Response 25: abbreviations were explained in Table 1 legend.

 Comment: 26) Results, add comments to support findings from Table 2, 3, and 4 in "Tables 2, 3, and 4 represent 78 QSAR models for bioactivities on S. aureus, E. coli, and C. Albicans."

Response 26: there are no 78 QSAR models

Reviewer's answer: Sorry for the mistake. It is not "78 QSAR models" but "QSAR models". Nevertheless, some comments are needed. '

Response 26: the following text was added: Using molecular descriptors and ligand-receptor docking data (regarding ATP synthase), QSAR models were developed. The five best-fit QSAR models are shown. For each model, comments were made regarding the QSAR model ( Table 2,3,4,)

Comment: 27) Results, p3: Explain what are the 5 descriptors, why S. aureus was selected, it was unclear if the target enzyme was the ATP synthase to support "Multiple linear regression (MLR) QSAR model for S. aureus was build using the 5 descriptors set (see Table 2) and shows a Pearson correlation (r)= 0.972, Pearson correlation squared (r2)= 0.945704, Spearman rank correlation (p)= 0.964286, Mean squared devi- 84 ation (MSD)= 1.38611, Root mean square deviation (RMSD)= 1.17733 and Cross validated 85 square (q2)= 0.945704 respectively. The MLR equation is y= 0.9457x+0.8995 (Figure 1)."

Comment 27: corrected. The following text was added : ( targeting ATP synthase )was built using the five descriptors set ( steric energy- steric energy of the compound kcal/mol; H bond -number of hydrogen bonds in the compound; heavy atoms-number of heavy atoms; N-number of nitrogen atoms, UnH bond- unbonded Hydrogen atoms;see Table 2)

Reviewer's answer: Indicate if S. aureus ATP synthase was used to build the five descriptors (PDB accessing number and reference for S. aureus ATP synthase are needed).

Response 27: AF_ AF_AFP99112F1 added as a reference and in text.

Comment: 28) Results, p3: It was unclear how the percentage of the phenomena was computed and to which phenomena it referred. Justify, add explanations to support "The QSAR model shows that nitrogen atoms and unidirectional hydrogen bonds have a crucial role in the activity of S. aureus, explaining 69% of the phenomena. A heavy atom's presence significantly increases the bioactivity of the compound (r2= 0.90). Steric and hydrogen bond energy can explain less than 5 % of phenomena."

Answer 28: corrected as suggested

Reviewer's answer: It is still unclear if the phenomena is directly associated to the ATP synthase activity.

Response 28: The descriptors used to compute the QSAR model are closely related to the ATP synthase ligan interaction (the algorithm selected descriptors used tu build the QSAR model) . Those descriptors correlate closely with the phenomena ( torsion angles, h bounding, halogen atoms). Docking studies were performed on S. Aureus  ATP synthase so some correlation must exist ( correlation demonstrated by the model itself)   

Comment: 29) Results, p3: The sentence was unclear. It was unclear what are the ligands to support "Ligand composition is imperative inactivity on S. aureus, while ligand conformation and bonding do not significantly impact."

Response 29: the sentence was corrected

Reviewer's answer: It is still unclear what the ligands are: the entire data set of ligands was added as supplinmetsl material S1.

Response 29: Scheme1

Comment: 30) Results, p3: Indicate why E.Coli was selected and if the enzyme target was the ATP synthase to support "ultiple linear regression (MLR) QSAR model for E. coli build using 5 descriptors (see Table 3) shows a Pearson correlation (r)= 0.947, Pearson correlation squared (r2)= 0.896, Spearman rank correlation (p)= 0.896, Mean squared deviation (MSD)= 2.039, Root mean square deviation (RMSD)= 1.428 and Cross validated square (q2)= 0.896 respectively. The MLR equation is y= 0.8968x+1.3927 (Figure 1)."

Response 30: corrected

Reviewer's answer: It is still unclear Indicate if E. coli ATP synthase was used to build the five descriptors (PDB  accessing number and reference for E. coli ATP synthase are needed).

Response 30:PD ID 6OQU

Sobti, M., Walshe, J.L., Wu, D., Ishmukhametov, R., Zeng, Y.C., Robinson, C.V., Berry, R.M., Stewart, A.G. Cryo-EM structures provide insight into how E. coli F1FoATP synthase accommodates symmetry mismatch. (2020) Nat Commun 11: 2615-2615

 Comment: 31) Results, p3: Specify the ligand to support "The QSAR model suggests that the ligand structure has a critical role in the activity of E. coli."

Response 31: ligand structures are shown in Scheme 1

Reviewer's answer. Add this information in the manuscript.

Response 31:            added.

Comment: 32) Results, p3: Add explanations to support "The degrees of freedom (DOF) 100 and N atoms can explain 70% of BQS3 bioactivity in E. coli. Also, the torsional energy of 101 the ligand halogen atom and unidirectional hydrogen bonds has been implied."

Respnse 32: corrected

Reviewer's answer. Add more explanations. 

 Response 32:

CommeRespont: 33) Results, p4: Indicate why C. Albicans was selected and if the enzyme target was the ATP synthasense to support "Multiple linear regression (MLR) QSAR model for C. Albicans build using 5 descriptors (Table 4) shows a Pearson correlation (r)= 0.9867, Pearson correlation squared (r2)= 0.9737, Spearman rank correlation (p)= 0.964, Mean squared deviation (MSD)= 0.548, Root mean square deviation (RMSD)= 0.740 and Cross validated square (q2)= 0.975 respectively. The MLR equation is y= 0.973x+0.4678 (Figure 1)."

Response 33: correted

Reviewer's answer: It is still unclear Indicate if C. Albicans ATP synthase was used to build the five descriptors (PDB  accessing number and reference for C. Albicans ATP synthase are needed).

Response 33: UniprotID  A0A1D8PDC4 strmg 237561.A0A1D8PDC4

 Comment: 34) Results, Fig 1, p4: Indicate in the Figure-1 legend what are the observed and predicted values to support "Figure 1. Scatter plot of observed (on ox axis) and predicted (on oy axis) MIC for: a. S. aureus; b. E. 115 coli, and c. C. albicans.”

Response 34: corrected

Reviewer's answer: It is insufficient. It is unclear if the minimum inhibitory concentrations referred to the ATP synthase activities or to antimicrobial activities.

Response 35: The minimum inhibitory concentrations refer to the antimicrobial activities. They added it to the figure 1 legend.

Comment: 35) Results, p4: It was unclear which activities and how predicted activities can be obtained to support "In Figure 1, scatter plots represent the observed and predicted activities on S. aureus, E. coli, and C. Albicans. Observed and predicted activities are roughly following each other. One point is an outlier for S. aureus and E. coli due to lack of activity."

Response 35: The QSAR methodology was described in the Material and methods section. The activities and predicted values have a correlation coefficient bigger than 0.9

Reviewer's answer: It is insufficient. It is unclear if the minimum inhibitory concentrations referred to the ATP synthase activities or to antimicrobial activities.

Response 35 : It refers to antimicrobial activity. It was corrected in the text.

Comment: 37) Results, p5: The sentence was too broad. Specify types of activity and Justify "Screening results retrieved many heterocyclic compounds with nitrogen. These findings follow the computed QSAR model and suggest exploring a distinct class of compounds with superior activity on discussed strains."

Response 37: the following text was added: As stated before, the screening template used was the pharmacophore hypothesis developed with the help of the QSAR model.

Reviewer's answer: The type of activity remained unclear. For the sake of clarity: Is that ATP synthase activity or is that antimicrobial activity ? No comparisons can be performed with antimicrobial activity since it may not be associated to a ATP synthase activity. The drug may act on a distinct target than ATP synthase if a microbial activity is measured.

Response 37: The study refers to antimicrobial activity as a drug targeting the ATP synthase (based on accepted data described in the literature). Regarding this class of compounds (benzoquinone) their mode if action is pretty clear.  

Comment: 38) Results, Figure 2, p6: Indicate the specie origin of the ATP synthase to support the Figure-2 legend in "Virtual screening cluster of compounds for BQS salts in interaction with ATP synthase is represented as a scatter plot: x- and y-axis represent coordinates of each compound characterized by its properties: relative energy, Phase vector score, phase volume score (see supplemental files)."

Response 39: corrected

Reviewer's answer: Not answered. Avoid broad sentences, indicate clearly specie origin for the calculations used (PDB of each microbial specy).

Response 38: the ATP synthase species and their id were indicated in the manuscript as suggested in previous comments.

 Comment: 40) Results, p6: Specify activities in "Tables 7, 8, and 9 represent QSAR models for bioactivities on S. aureus, E. coli, and C. Albicans."

 Response 40: corrected

Reviewer's answer: Not answered. indicate clearly types of activities. Is that ATP synthase activity ?

Response 40: No is the antimicrobial activity in respect to TOPO II. The following text was added: Tables 7, 8, and 9 represent QSAR models for bioactivities considering TOPO II as a target on S. aureus, E. coli, and C. Albicans.

 Comment: 41) Results, p7: It was unclear to which type of ATP synthase the data from table 7 refers to in "Multiple linear regression (MLR) QSAR model for S. aureus was build using 5 descriptors (see Table 7) and shows a Pearson correlation (r)= 0.960, Pearson correlation squared (r2)= 0.921788, Spearman rank correlation (p)= 0.88, Mean squared deviation 149 (MSD)= 1.996, Root mean square deviation (RMSD)= 1.413 and Cross validated square 150 (q2)= 0.921788 respectively."

Response 41: corrected

Reviewer's answer: No comparisons can be performed with antimicrobial activity since it may not be associated to the ATP synthase activity. The drug may act on a distinct target than ATP synthase if a microbial activity is measured.

Response 41: Benzpquinoline target is demonstrated to be ATP synthase and TOPO II

Comment: 42) Results, p7: It was unclear which type of bioactivities and which phenomena to support "QSAR models show that the presence of an N atom can explain 42% of the phenomena. The degrees of freedom also have a crucial role in bioactivity. Pose energy and torsion energy count for the rest of the phenomena."

Response 42: corrected

Reviewer's answer: No comparisons can be performed with antimicrobial activity since it may not be associated to the ATP synthase activity. The drug may act on a distinct target than ATP synthase if a microbial activity is measured.

Response 42; Benzoquinolines concerning antimicrobial activity regarding S. aureus, E. coli, and C. Albicans. have two targets ATP synthase and TOPO II

Comment: 43) Results, p7: It was unclear to which type of ATP synthase the data from table 8 refers to in "Multiple linear regression (MLR) QSAR model for E. coli build using the 5 descriptors a Pearson correlation (r)= 0.92798, Pearson correlation squared (r2)= 0.882256, Spearman rank correlation (p)= 0.91785, Mean squared deviation (MSD)= 2.32741, Root mean square deviation (RMSD)= 1.52559 and Cross validated square (q2)= 0.882256 respectively. The MLR equation is y= 0.8823x+1.5895 (Figure 4)."

Response 43: corrected

Reviewer's answer: No comparisons can be performed with microbial activity since it may not be associated to the ATP synthase activity. The drug may act on a distinct target than ATP synthase if antimicrobial activity is measured.

 Response 43: Benzoquinolines in respect of antimicrobial activity regarding S. aureus, E. coli, and C. Albicans. have two targets, ATP synthase, and TOPO II

Comment: 45) Results, p7: It was unclear to which type of ATP synthase the data from table 9 refers to in "Multiple linear regression (MLR) QSAR model for C. albicans build using 5 descriptors (Table 9) shows a Pearson correlation (r)= 0.9867, Pearson correlation squared (r2)= 0.9737, Spearman rank correlation (p)= 0.91785, Mean squared deviation (MSD)= 2.32741, Root mean square deviation (RMSD)= 1.52559 and Cross validated square (q2)= 0.882256 respectively. The MLR equation is y= 0.9737x+1.5895 (Figure 4)."

Response 45: corrected

Reviewer's answer: No comparisons can be performed with microbial activity since it may not be associated to the ATP synthase activity. The drug may act on a distinct target than ATP synthase if a antimicrobial activity is measured.

Response 45: Benzoquinolines, in respect of antimicrobial activity regarding S. aureus, E. coli, and C. Albicans. have two targets, ATP synthase and TOPO II

Comment: 46) Results, p7: It was unclear how membrane permeabilities were assessed to support "QSAR model shows that bioactivity on C. Albicans of BQS salts three could be explained by the membrane permeability properties of the compounds. In addition, findings suggest that the polarizability of the ligand has a crucial role."

Response 46: membrane operability was assessed by using membrane permeability descriptors. The following text was added: characterized by using membrane permeability descriptors -see Table 9).

Reviewer's answer: It is still unclear how membrane permeabilities were determined. Table 9 apparently did not report any membrane permeability data.

Response 46: PEOE_VSA+2, PREOE_VSA-0, and PEOE_VSA_FPPOS are a descriptor related to membrane permeability.

Comment: 47) Results, Fig. 4, p8: Specify the observed and predicted values in the Figure-4 legend to support "Figure 4. Scatter plot of observed (on ox axis) and predicted (on oy axis) MIC for a. S. aureus; b. E. 177 coli, and c. C. albicans.”

Response 47: corrected

Comment: 48) Results, p8: Specify which type of bioactivity to support "Overall the scatter plot demonstrates a correct prediction of the variable (bioactivity) by using the multiple linear regression 181 (MLR) QSAR models."

Response 48: corrected

Reviewer's answer: Not answered. indicate clearly types of activities. Is that ATP synthase activity ?

 Response 48: It is the antimicrobial activity.

Comment: 50) Results, p8: Explain why topoisomerase II was selected and Indicate the specie origin of ATP synthase and Topoisomerase II. To support "Screening results show that compounds with heterocycles in structure and halogen 188 atoms bound with aromatic rings satisfy the hypothesis A1P2R3R4R5R6R7 and can be 189 active on ATP synthase and Topoisomerase II."

Answer 50: corrected. The following text was added: Both enzymes are crucial in bacterial grouth and multiplication, being preferable antibiotics targets.

Reviewer's answer: indicate if ATP synthase activity of Topoisomerase II was targeted ?

 Response 50: Both a common pharmacophore hypothesis was used.Tables 1 and 6 

Comment: 51) Discussion, p9: Indicate the specie origin of ATP synthase, and TOPO II. "Regarding screening compounds' structural properties, it is noted that both series of the compound screen against the two enemies, TOPO II and ATPase, have the same structural properties."

 Answerer 51: corrected

Reviewer's answer: indicate if ATP synthase activity of Topoisomerase II was targeted ?

 Rwesponse 51: both

Comment: 52) Discussion, Fig 7, p10: Indicate the specie origin of ATP synthase, and TOPO II to support Figure-7 legend in . "Structural composition of molecular dataset resulted after screening"

Answer 52: corrected

Reviewer's answer: It is unclear  if ATP synthase activity was determined.

Resonse 52: antimicrobial activity was determine

 Comment: 53) Materials and Methods, p11: Indicate the species of TOPO II and ATP-synthase "Using to support, indicate chemical structures of the two compounds to support "Two essential compounds synthesized and tested [8], two single pharmacophore hypotheses were developed for best salts compounds with topoisomerase II (TOPO II ) and Adenosine triphosphate synthase (ATP-synthase) as targets."

Response 53: corrected

Reviewer's answer: It remains unspecified. I have to dig from the previous papers of the authors to find an answer. From reference: Vasilichia Antoci et al.  Benzoquinoline Derivatives: A Straightforward and Efficient Route to Antibacterial and Antifungal Agents. Pharmaceuticals 2021, 14(4), 335; https://doi.org/10.3390/ph14040335, it appears that human V-ATPase and not bacterial ATPase was used for docking. The docking and theoretical predictions based from the human V-ATP synthase can't be used for microbial ATP synthase. Poisons targeting human ATP synthase  instead of drugs targeting microbial ATP synthase are being screened.

Response 54: bacterial species

Comment: 55) Materials and Methods, p11: Indicate the specie of the enzymes in "TOPO II and ATP -synthase. PDB structures of TOPO II and ATP synthase were used."

Reviewer's answer: It remains unspecified. See comment 53.

 Response 55: bacterial species

Comment: 56) Conclusions, p12: It was unclear if compounds shall target the microbial enzymes or the human enzymes since there were insufficient information how biological activities were measerd. Rephrase, add information to support "Screening results show that compounds with heterocycles in structure and halogen atoms bound with aromatic rings satisfy the hypothesis A1P2R3R4R5R6R7 and can be active on ATP synthase and Topoisomerase II. Furthermore, antimicrobial and antitumoral activity seems to be related to the lipophilic properties of the compounds."

Response 56: corrected.

Reviewer's answer: It remains unspecified. See comment 53.

Response 56: Corrections were performed in the manuscript as suggested.

Reviewer 2 Report

The authors have taken great care to improve the manuscript taking into account the reviewers' comments, and I can now recommend the manuscript for publication.

Author Response

The introductions, methods, and conclusion have been improved. 

Thank you for reviewing our manuscript.